# FINE-GRAINED URBAN TRAFFIC FORECASTING ON METROPOLIS-SCALE ROAD NETWORKS

## ABSTRACT

Traffic forecasting on road networks is a complex task of significant practical importance that has recently attracted considerable attention from the machine learning community, with spatiotemporal graph neural networks (GNNs) becoming the most popular approach. The proper evaluation of traffic forecasting methods requires realistic datasets, but current publicly available benchmarks have significant drawbacks, including the absence of information about road connectivity for road graph construction, limited information about road properties, and a relatively small number of road segments that falls short of real-world applications. Further, current datasets mostly contain information about intercity highways with sparsely located sensors, while city road networks arguably present a more challenging forecasting task due to much denser roads and more complex urban traffic patterns. In this work, we provide a more complete, realistic, and challenging benchmark for traffic forecasting by releasing datasets representing the road networks of two major cities, with the largest containing almost 100,000 road segments (more than a 10-fold increase relative to existing datasets). Our datasets contain rich road features and provide fine-grained data about both traffic volume and traffic speed, allowing for building more holistic traffic forecasting systems. We show that most current implementations of neural spatiotemporal models for traffic forecasting have problems scaling to datasets of our size. To overcome this issue, we propose an alternative approach to neural traffic forecasting that uses a GNN without a dedicated module for temporal sequence processing, thus achieving much better scalability, while also demonstrating stronger forecasting performance. We hope our datasets and modeling insights will serve as a valuable resource for research in traffic forecasting and, more generally, urban computing and smart city development.

## 1 INTRODUCTION

Accurate traffic forecasting on road networks is a critical task with significant practical implications for urban planning, logistics optimization, and the daily experience of commuters (Li et al., 2018; Derrow-Pinion et al., 2021; Lim & Zohren, 2021; Jiang & Luo, 2022). In recent years, substantial efforts from the machine learning community have been dedicated to this challenge, with spatiotemporal graph neural networks (GNNs) emerging as the dominant methodology due to their inherent ability to model complex spatial and temporal dependencies (Cini et al., 2023).

However, the development and proper evaluation of advanced traffic forecasting methods depend critically on the availability of realistic and comprehensive benchmarks. Unfortunately, current publicly available traffic datasets have significant drawbacks that hinder progress in the field. In the existing traffic forecasting benchmarks (Jagadish et al., 2014; Li et al., 2018; Yu et al., 2018; Guo et al., 2019; Song et al., 2020; Liu et al., 2023), nodes represent sensors located on roads that measure traffic speed, and edges are constructed based on location proximity (road travel distance between the sensors). These sensors are sparsely distributed and are mostly located on intercity highways, which leads to a number of limitations. First, the overall number of locations (road segments) with available measurements is relatively small, ranging from 207 to 8,600 in the largest currently available dataset. Second, there is no graph structure available between the sensors due to their sparsely distributed locations. Thus, in the existing datasets, graph edges are heuristically constructed based on the road distances, leaving the natural graph structure arising from adjacent road segments underexplored. Finally, since sensors are typically located on intercity highways, their measurements fail to capture

complex urban traffic within cities, which is a critical limitation, since traffic conditions within large cities affect daily commutes of millions of people.

To address these problems, our work provides a realistic and challenging benchmark specifically tailored for urban traffic forecasting. We release novel datasets representing the detailed road networks of two major cities. The largest of these datasets encompasses information for almost 100,000 distinct road segments of a major city of approximately 5.5 million residents. Our datasets contain rich road features and provide fine-grained temporal data capturing both traffic volume and traffic speed, enabling the development and evaluation of more holistic and nuanced traffic forecasting systems.

Using our datasets, we examine several existing implementations of neural traffic forecasting models and show that most of them struggle to scale to data of this magnitude. To overcome this issue, we propose an efficient approach to neural traffic forecasting that uses a GNN without a dedicated module for temporal sequence processing, thus achieving much better scalability, while also demonstrating stronger forecasting performance.

We hope our proposed datasets and modeling insights will stimulate further advancements in traffic forecasting and, more broadly, support progress in the related fields of urban computing and smart city development.

## 2 BACKGROUND

### 2.1 TRAFFIC FORECASTING

The goal of traffic forecasting is to predict future traffic conditions (e.g., traffic speed or traffic volume) based on historical observations. Typically, observations are provided by sensors located at specific road segments. Traditional approaches that rely on statistical models, such as ARIMA or Kalman filters, often fall short in capturing the complex, nonlinear spatiotemporal dependencies present in real-world traffic systems. Recent advances in deep learning, particularly in representation learning on graphs and sequences, have led to a surge of interest in neural methods for traffic forecasting, aiming to model spatial and temporal components jointly and more effectively.

One of the pioneering works in this direction is Diffusion Convolutional Recurrent Neural Network (DCRNN) proposed by Li et al. (2018), which formulates the traffic forecasting problem as a spatiotemporal sequence modeling task, representing the traffic network as a directed graph and utilizing diffusion convolution over the graph structure to capture spatial dependencies, integrated with a recurrent neural network (RNN) to model the temporal component. This work was one of the first to use graph-based convolutions in traffic forecasting, so it became the groundwork for many subsequent methods.

Building on this, Yu et al. (2018) proposed Spatiotemporal Graph Convolutional Network (STGCN) that replaces RNNs with temporal convolutional layers, resulting in improved computational efficiency. This architecture employs separate modules for spatial and temporal components, alternating between graph convolutional networks (GCNs) for aggregating local structural information and 1D convolutions for processing sequential information.

Later works sought to address the limitations of previous models by introducing more intricate and flexible mechanisms. For instance, Attention-based Spatial-Temporal Graph Convolutional Networks (ASTGCN) by Guo et al. (2019) incorporate spatial and temporal attention to dynamically weigh the importance of different nodes and time steps, potentially improving the model's ability to focus on specific patterns. Similarly, Graph WaveNet by Wu et al. (2019) introduces adaptive adjacency matrices and dilated temporal convolutions to enable the model to learn latent spatial structure and long-range temporal dependencies more efficiently. Another work in this direction is Adaptive Graph Convolutional Recurrent Network (AGCRN) by Bai et al. (2020) that learns node embeddings and constructs adaptive graphs dynamically, decoupling model performance from reliance on predefined graph structures.

Further, Zheng et al. (2020) introduce a fully attention-based architecture in Graph Multi-Attention Network (GMAN), avoiding both recurrent and convolutional components, and combining spatial and temporal attention to dynamically model the spatiotemporal patterns at each time step. Together with other examples, such as Dynamic Spatial-Temporal Aware Graph Neural Network (DSTAGNN) by Lan et al. (2022) and Spatial-Temporal Transformer Networks (STTNs) from Xu et al. (2020),

these works mark a trend in the field towards attention-based models and even more sophisticated methods for capturing complex dependencies in the data.

As can be seen, many recent models incorporate multiple complex components, such as hierarchical attention or adaptive adjacency learning, which can significantly complicate implementation and introduce overheads in computation. Consequently, scaling to large traffic networks with tens of thousands of sensors can become a great challenge for these models, since implementing and training them efficiently is a non-trivial task, and the real-time deployment of such models can be hindered by their computational complexity.

For most of the discussed models, there are publicly available implementations that have been introduced by the authors of the original works or provided by the authors of existing traffic forecasting benchmarks (Liu et al., 2023). However, as we discuss further, the currently available traffic datasets do not allow us to thoroughly evaluate these implementations and ensure their practical usability for large-scale traffic forecasting, since they do not provide enough road feature information and complete city road network to test the performance of traffic forecasting models.

## 2.2 LIMITATIONS OF EXISTING DATASETS

By far the most popular datasets for traffic forecasting are `METR-LA` and `PEMS-BAY` introduced by Li et al. (2018). In these datasets, nodes represent sensors located on roads that measure traffic speed, and edges are constructed based on location proximity (road travel distance between the sensors). `METR-LA` is based on data from loop detectors in the highways of Los Angeles County (Jagadish et al., 2014) and `PEMS-BAY` is based on data from California Department of Transportation (CalTrans) Performance Measurement System (PeMS, Chen et al., 2001). Some works also use other datasets collected from the same PeMS data source: these datasets may include different subsets of sensors or measurements during different periods of time, but the general structure of these datasets is mostly the same (Yu et al., 2018; Guo et al., 2019; Song et al., 2020). Most works on GNN-based traffic forecasting evaluate their models exclusively on `METR-LA`, `PEMS-BAY`, or other datasets obtained from the PeMS data.

We note that these standard datasets are extremely small: `METR-LA` has only 207 nodes (sensors), while `PEMS-BAY` has only 325 nodes. Other traffic forecasting datasets obtained from the PeMS data also typically have up to a few hundred nodes. Recently, a larger dataset based on PeMS data was proposed: `LargeST` (Liu et al., 2023) with 8,600 nodes, which is still relatively small compared to the amount of data that needs to be processed by traffic forecasting systems in large cities. The small size of standard datasets de-emphasizes model efficiency and leads to proposed models being very resource-intensive and thus not scalable to real-world applications, as we will discuss later.

To obtain a graph structure, previous works (Li et al., 2018; Yu et al., 2018; Wu et al., 2019; Liu et al., 2023) connect two sensors if the road network distance between them is below a certain heuristically chosen threshold. The real road graph structure cannot be used, since sensors are sparsely located. Thus, the only option is to use a heuristic for constructing a graph in the absence of information about the network connectivity. As a result, the real network connectivity is not provided with any of the standard datasets, which is a significant limitation.

Further, in all currently available traffic forecasting datasets, sensors (graph nodes) are sparsely distributed and only cover a relatively small number of roads. We provide the visualizations of the geographic distribution of sensors in `METR-LA`, `PEMS-BAY`, and `LargeST` datasets in Figure 1. It can be seen that sensors in these datasets are sparse and most of them have only two direct neighbors in the road graph (the sensors right before and after on the same road), with only a small share of sensors located near intersections. This limits the possibility of using these datasets to study complex traffic patterns. The reason for this is that these datasets mostly focus on large interstate highways which, despite having numerous sensors, feature sparse sensor distribution. At the same time, city streets are almost not represented in these datasets. However, urban traffic is arguably more complex, presents unique patterns, and is more challenging to forecast.

## 3 NEW `city-traffic` DATASETS

In our work, we present the first openly available datasets for large-scale and fine-grained study of urban traffic. We collect two spatiotemporal graph datasets from two major cities: `city-traffic-M`

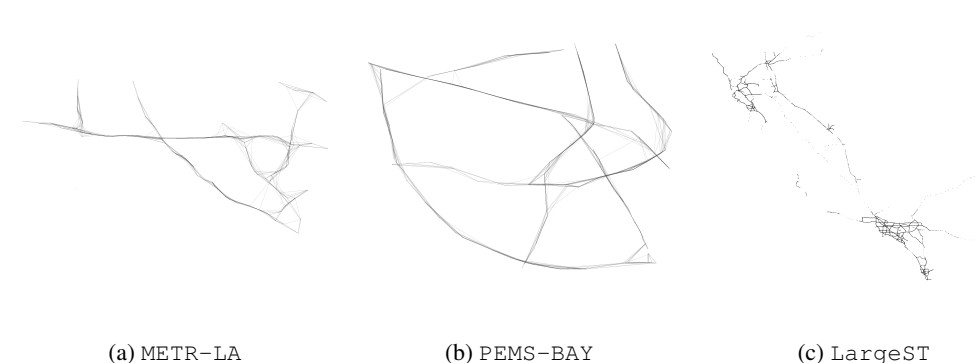

(a) `METR-LA`       (b) `PEMS-BAY`       (c) `LargeST`

Figure 1: Visualization of existing traffic forecasting datasets. Nodes correspond to sensors; graph structure is heuristically constructed based on road distances; layout is defined by sensor locations.

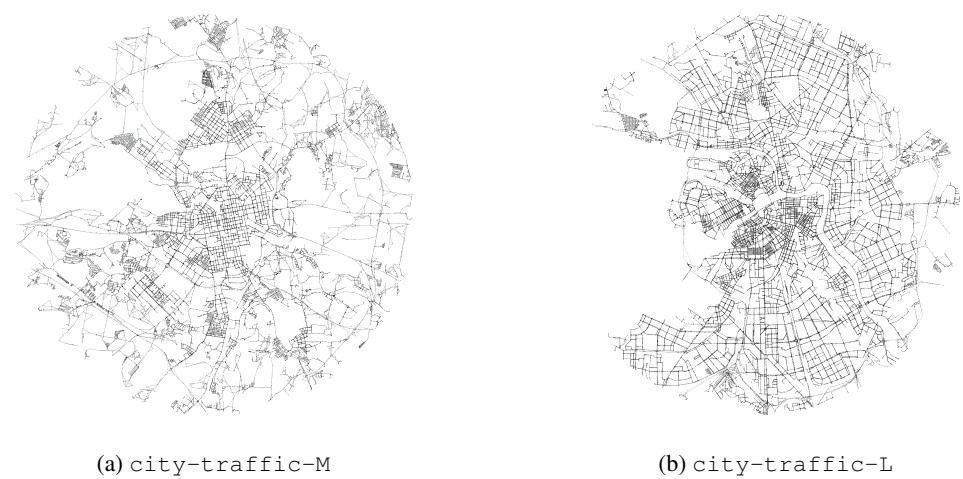

(a) `city-traffic-M`       (b) `city-traffic-L`

Figure 2: Visualization of the proposed datasets. Nodes correspond to road segments; graph structure is defined by road adjacency; layout is defined by segment locations.

with more than $50,000$ nodes and `city-traffic-L` with almost $100,000$ nodes. These datasets differ significantly from the previous traffic forecasting datasets in what the graphs represent and how they are constructed. While previous datasets only have information about traffic at the locations of sensors, which are only placed at some roads and are generally sparse, the information in our datasets was obtained from GPS measurements rather than sensors, and therefore the measurements are available at a fine-grained level of individual road segments. Thus, our graphs have nodes corresponding to *all road segments in the two considered cities*. Further, while previous datasets construct edges heuristically based on travel distance between sensors, our graph has edges representing actual road connectivity, which can provide much more information. In our graphs, a directed edge connects two road segments if they are incident to each other and moving from one segment to the other is permitted by traffic rules. Next, our datasets have rich node features describing the properties of road segments, including speed limits — important information absent from all widely used traffic forecasting datasets. Our datasets are also the first providing information on traffic volume and traffic speed simultaneously, allowing for a more holistic approach to traffic forecasting. Thus, our datasets represent a realistic setting of traffic forecasting by a traffic monitoring system, which contrasts with the previous datasets that only roughly approximate it due to incomplete data. Some characteristics of our and existing datasets are shown in Table 1.

What makes our datasets fundamentally different from the currently widely used ones is that they focus on urban traffic with its high road density and complex patterns and dynamics. We provide

Table 1: Dataset characteristics

| dataset | speed | volume | # roads | # road attributes | real connectivity | reference |
|---------|:-----:|:------:|--------:|:-----------------:|:-----------------:|-----------|
| `PeMSD7(M)` | ✓ | ✗ | 228 | 6 | ✗ | Yu et al. (2018) |
| `PeMSD7(L)` | ✓ | ✗ | 1,026 | 0 | ✗ | |
| `METR-LA` | ✓ | ✗ | 207 | 3 | ✗ | Li et al. (2018) |
| `PEMS-BAY` | ✓ | ✗ | 325 | 3 | ✗ | |
| `PEMS03` | ✗ | ✓ | 358 | 1 | ✗ | |
| `PEMS04` | ✗ | ✓ | 307 | 0 | ✗ | Song et al. (2020) |
| `PEMS07` | ✗ | ✓ | 883 | 0 | ✗ | |
| `PEMS08` | ✗ | ✓ | 170 | 0 | ✗ | |
| `LargeST` | ✗ | ✓ | 8,600 | 9 | ✗ | Liu et al. (2023) |
| `city-traffic-M` | ✓ | ✓ | 53,530 | 26 | ✓ | ours |
| `city-traffic-L` | ✓ | ✓ | 94,009 | 26 | ✓ | |

Figure 3: The weekly dynamics of target variables averaged across all roads in the proposed datasets.

the visualizations of our datasets in Figure 2. It can be seen that our road networks are much more interconnected and present more complex structural patterns than in the previous datasets.

For each road segment, we provide two dynamic variables: current traffic speed and volume, both estimated using high-resolution GPS signals transmitted by vehicles. This data is provided at a 5-minute granularity, spanning from July 1st, 2024, to November 1st, 2024. For speed, missing values can occur due to insufficient GPS signals for certain road segments at specific timestamps. For example, in `city-traffic-L`, the proportion of missing speed values can range from 5% to 25%, depending on the time of day — a level of missingness consistent with real traffic data. For traffic volume, there are no missing values.

Finally, for each road segment, we provide 26 static attributes that describe various properties of the segment, including its length, speed limit, coordinates of the segments' endpoints, quality of road surface, indicator to masstransit lane, presence of crosswalk, restriction for certain types of vehicles, and so on. Road attributes are a mixture of numerical, categorical, and binary features. More detailed information about the proposed datasets can be found in Appendix A.

In Figure 3, we visualize the behavior of dynamic target variables: traffic volume and speed. For each variable and each city, we average the values over all road segments in the city. One can clearly see the daily traffic patterns — e.g., there are noticeable traffic jams in the morning and evening on working days, which are indicated by the rapid decrease of average traffic speed and increase of traffic volume. The same target variables change more gradually and have smaller variance on holidays. While the average speed in `city-traffic-M` and `city-traffic-L` is similar, traffic volumes differ significantly. Figure 4 also provides the distribution of traffic volume and speed for each of the

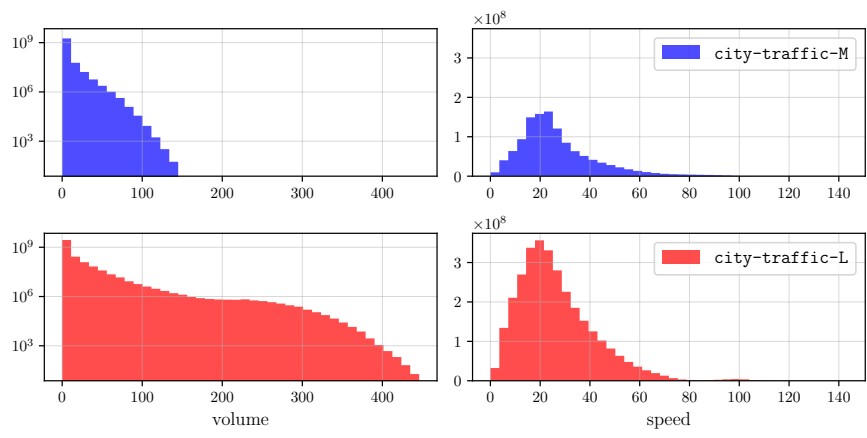

Figure 4: The histograms of traffic volume and speed in the proposed datasets.

datasets. It can be seen that the amount of traffic varies significantly across the considered cities. A more in-depth discussion of the difference between the datasets can be found in Appendix B and C.

## 4 EXPERIMENTS

In this section, we evaluate the scalability and forecasting performance of existing neural spatiotemporal models on our fine-grained traffic datasets. We benchmark several established architectures and highlight their limitations in handling datasets of our size and complexity. To address these limitations, we then introduce a simple but effective model that scales well to large dataset sizes and also outperforms existing baselines in forecasting accuracy. We describe the experimental setup in Appendix D.

### 4.1 MODELS

**Simple graph-agnostic baselines**  First, we evaluate several naive baselines to establish reference points for model performance. These baselines rely on simple heuristics derived from past traffic values. The simplest of these baselines is the *previous* strategy, which predicts the most recently observed value at each road segment. We also consider baselines that use the daily and weekly periodicity in traffic patterns, which is commonly observed in urban traffic dynamics. Namely, we predict traffic speed/volume by using the corresponding value either one day or one week ago from the target timestamp. We refer to these methods as *previous 1 day/week ago*. Next, we include simple statistical baselines such as the global *mean*, *median*, as well as *node-wise mean* and *node-wise median* which are the mean and median computed independently for each road segment. These naive baselines do not exploit the graph structure. We also evaluate a linear model that can be considered as a basic graph-agnostic baseline.

**Spatiotemporal baselines**  For our experiments, we have selected four popular models from the literature that are frequently used by other works on graph-based time series forecasting and that could scale to our datasets (see below). To process the temporal dimension of the data, they utilize either recurrence or convolution mechanisms:

- DCRNN (Li et al., 2018) — a diffusion convolutional recurrent neural network that exploits recurrent cells supplied with a graph convolution operation;

- GRUGCN (Gao & Ribeiro, 2022) — a combination of recurrent temporal encoder and graph convolutional spatial encoder, which are stacked consecutively;

- STGCN (Yu et al., 2018) — a spatiotemporal graph neural network that is composed of alternating temporal and graph convolution operations;

- GWN (Wu et al., 2019) — a spatiotemporal graph neural network that stacks graph convolutions and causal dilated temporal convolutions.

For our experiments, we adapt the implementations from the `LargeST` repository (Liu et al., 2023).

**Scalable traffic forecasting approach** Our datasets are much larger than the ones currently used in the literature. Thus, they present a significant scaling challenge to deep learning models. We investigated the models available in Torch Spatiotemporal (Cini & Marisca, 2022) as well as in the codebase of LargeST (Liu et al., 2023), the largest previous traffic forecasting dataset, and found that **only four models listed above** can be trained on city-traffic-M on a GPU with 80GB VRAM. However, even these models require very long training time. This led us to investigate the sources of the inefficiency of the currently available methods and look for ways to design more scalable models.

The GNN-based models for traffic forecasting proposed in previous works typically use recurrence, convolution, or attention mechanisms to process the temporal dimension of the data. However, these mechanisms are relatively resource-intensive since they maintain a separate vector representation for each timestamp in the lookback window for each node in the graph. Thus, for a dataset with $n$ graph nodes, a lookback window of $t$ timestamps, and a hidden dimension of size $d$, each layer of such models requires at least $\mathcal{O}(ntd)$ memory. While the aforementioned mechanisms differ in their required number of operations (and their ability to parallelize them), for all of them it is at least linear in the number of vector representations, which is $\mathcal{O}(nt)$, and each of these representations is involved in at least one matrix-vector multiplication, so each layer also performs at least $\mathcal{O}(ntd^2)$ operations. Thus, for datasets with a large number of nodes or a necessity to use a long lookback window, the time and memory requirements of such models quickly become prohibitive.

However, in the time series literature, several recent works have been exploring an alternative direction that allows processing the temporal dimension much more efficiently (Oreshkin et al., 2019; Zeng et al., 2023; Zhang et al., 2022; Das et al., 2023; Li et al., 2023; Yi et al., 2024). These works concatenate all past time series values in the lookback window into a single input vector and transform it into a single vector representation (e.g., with one linear layer). This vector representation is then processed with an MLP-based model (Zeng et al. (2023) do not use an MLP at all and directly make predictions with just one linear layer). Despite the simplicity of this approach, it has been shown that it can compete with other models or even outperform them, all while being significantly, sometimes orders of magnitude, more efficient.

In this work, we propose to exploit this approach for graph-based traffic forecasting. Specifically, we take the idea of encoding each time series in a multivariate dataset into a single vector representation with a linear layer and adapt it to graph-based forecasting setting by replacing the following MLP with a GNN. Since, crucially, this approach requires maintaining only a single vector representation per graph node (in contrast to $t$ vector representations required by other methods), in the case of graph-based traffic forecasting, it has per-layer memory complexity of only $\mathcal{O}(nd)$, which allows it to efficiently scale to much larger datasets, such as the ones we propose in our work.

Our proposed model consists of a linear layer that encodes the temporal information of a single time series into a latent vector representation and a multilayer GNN that allows representations of different time series to interact according to the graph connectivity. According to the categorization of temporal graph models introduced by Gao & Ribeiro (2022), models sharing our approach are *time-then-graph* models (in contrast to more popular *time-and-graph* models), but their component for processing the temporal dimension is extremely simplified (e.g., to a single linear layer) for the purpose of efficiency.

Our approach can use any GNN architecture. For our experiments, we use GNNs with two popular spatial graph convolution mechanisms: mean aggregation, which was popularized in modern GNNs by Hamilton et al. (2017), and transformer-like multihead attention aggregation that has been popularized in GNNs by Shi et al. (2021) (note that this is attention over graph neighbors, not global attention). We refer to these models as GNN-Mean and GNN-TrfAttn. Following Platonov et al. (2023); Bazhenov et al. (2025), we augment our GNNs with skip connections (He et al., 2016), layer normalization (Ba et al., 2016), and MLP blocks, which often significantly improve their performance.

We show that our approach, despite its simplicity and efficiency, often leads to better forecasting quality than prior methods. We also show that its efficiency allows it to use much longer lookback windows with a negligible impact on computational cost (since only a single linear layer is affected), which often further improves the forecasting performance. We hope that these findings will encourage further development of efficient methods for traffic modeling and graph-based spatiotemporal forecasting in general.

Table 2: Performance of simple baselines and spatiotemporal models, MAE on the test set is reported. `MLE` indicates setups which did not fit in GPU memory.

| | | city-traffic-L | | city-traffic-M | |
| --- | --- | --- | --- | --- | --- |
| | | volume | speed | volume | speed |
| naive baselines | mean | 9.4130 | 11.8283 | 2.8480 | 11.7040 |
| | median | 7.5765 | 11.5509 | 2.0627 | 11.1612 |
| | node-wise mean | 5.3547 | 5.9122 | 1.5265 | 5.4479 |
| | node-wise median | 5.2967 | 5.8183 | 1.4913 | 5.3751 |
| | previous | 2.6405 | 4.5764 | 0.9567 | 4.2404 |
| | previous 1 day ago | 2.8076 | 5.8273 | 0.9876 | 5.5497 |
| | previous 1 week ago | 2.5396 | 5.6997 | 0.9264 | 5.4758 |
| | Linear model | $2.284 \pm 0.000$ | $4.229 \pm 0.001$ | $0.806 \pm 0.000$ | $3.951 \pm 0.001$ |
| spatiotemporal | DCRNN | $2.212 \pm 0.054$ | $3.988 \pm 0.012$ | $0.765 \pm 0.007$ | $3.704 \pm 0.014$ |
| | GRUGCN | $2.255 \pm 0.011$ | $4.074 \pm 0.014$ | $0.765 \pm 0.011$ | $3.717 \pm 0.020$ |
| | STGCN | MLE | MLE | $0.777 \pm 0.011$ | $3.663 \pm 0.016$ |
| | GWN | $2.368 \pm 0.006$ | $4.516 \pm 0.008$ | $0.792 \pm 0.004$ | $4.204 \pm 0.083$ |
| | GNN-Mean | $2.038 \pm 0.021$ | $3.753 \pm 0.005$ | $0.737 \pm 0.004$ | $3.397 \pm 0.011$ |
| | GNN-TrfAttn | $2.050 \pm 0.029$ | $3.724 \pm 0.010$ | $0.733 \pm 0.006$ | $3.353 \pm 0.007$ |

## 4.2 RESULTS

**Model comparison**    First, we compare the performance of the considered models; the results are shown in Table 2. Following previous studies, we use the lookback window of 12. Among the considered naive baselines, the best results for traffic volume prediction are achieved by the predictor taking the value one week ago from the target timestamp; for speed prediction, the best naive predictor employs the latest known value. These metric values should serve as a necessary sanity check to ensure that the designed models actually capture useful information for the given forecasting task. Thus, as expected, the linear model consistently outperforms the presented naive baselines, which demonstrates that using historic observations is essential for precise traffic forecasting. More advanced spatiotemporal methods, in turn, have better performance than all graph-agnostic approaches, which indicates that using structural information about the road network is important for accurate traffic forecasting. Among the considered graph-aware methods, the best results are almost always achieved by the proposed GNN-TrfAttn model. These results suggest that models with more flexible mechanism for aggregating structural information, such as Transformer self-attention, have more potential for generalizing to complex traffic networks, so they should be especially considered when developing more effective backbones for spatiotemporal traffic forecasting.

**Effect of lookback window**    In the next series of experiments, we vary the lookback window among the following options: $[12, 24, 36, 48, 72]$ and consider the best-performing and efficient model GNN-TrfAttn with 2 layers and 512 hidden dimension size. As can be seen from Table 3, better results can usually be achieved for larger lookback windows, which proves that more complete information about how the target variable changed in the past is important for more accurate predictions in the future. At the same time, these results show that even such a simple module for processing the temporal component as a linear projection of historical variables into latent space of GNN model allows it to scale to greater amounts of data, while preserving computational efficiency.

**Scalability of the models**    We report the total training time in hours for all evaluated models across different datasets and lookback window sizes of 12 and 48 in Table 4. As the lookback window increases from 12 to 48, the considered sequential models, especially DCRNN, GWN, and STGCN, exhibit significantly worse scaling behavior. In case of STGCN on city-traffic-L dataset with a lookback of 48, training fails to complete within 250 hours. This poor scalability is attributed to the need to maintain and process an explicit temporal state for each input timestamp, which grows linearly with the lookback size. In contrast, our proposed models GNN-Mean and GNN-TrfAttn require consistently low training time across all configurations. This demonstrates that such a non-sequential full-batch design is significantly more scalable and computationally efficient, particularly as the

Table 3: Effect of lookback horizon on model performance, MAE on the test set is reported.

| | lookback | city-traffic-L volume | city-traffic-L speed | city-traffic-M volume | city-traffic-M speed |
|---|---|---|---|---|---|
| GNN-TrfAttn | 12 | $2.042 \pm 0.027$ | $3.818 \pm 0.004$ | $0.748 \pm 0.008$ | $3.504 \pm 0.010$ |
| | 24 | $2.033 \pm 0.026$ | $3.778 \pm 0.006$ | $0.751 \pm 0.007$ | $3.457 \pm 0.013$ |
| | 36 | $2.017 \pm 0.010$ | $3.773 \pm 0.015$ | $0.744 \pm 0.009$ | $3.431 \pm 0.010$ |
| | 48 | $2.021 \pm 0.021$ | $3.761 \pm 0.000$ | $0.743 \pm 0.005$ | $3.428 \pm 0.008$ |
| | 72 | $2.021 \pm 0.016$ | $3.743 \pm 0.002$ | $0.743 \pm 0.013$ | $3.414 \pm 0.009$ |

Table 4: Training time in hours for different models across two datasets and two lookback window sizes. TLE indicates models that did not converge within a 250 hours time limit.

| | city-traffic-L | | city-traffic-M | |
|---|---|---|---|---|
| Lookback | 12 | 48 | 12 | 48 |
| DCRNN | 7.52 | 31.17 | 5.13 | 21.06 |
| GRUGCN | 2.24 | 7.63 | 1.24 | 4.12 |
| STGCN | 26.55 | TLE | 6.38 | 211.19 |
| GWN | 6.84 | 27.81 | 4.17 | 17.13 |
| GNN-Mean | 1.45 | 1.79 | 0.77 | 0.90 |
| GNN-TrfAttn | 1.88 | 2.09 | 1.06 | 1.18 |

temporal input dimension grows. These results highlight the importance of scalability aspect for practical application of traffic forecasting models.

## 5    DISCUSSION & FUTURE OPPORTUNITIES

Our work makes a twofold contribution to the field of traffic forecasting. First, we introduce two novel large-scale datasets for fine-grained urban traffic forecasting: city-traffic-M and city-traffic-L. These datasets address critical limitations of existing benchmarks by providing the detailed coverage of urban road segments rather than sparse sensor locations; actual road network connectivity instead of heuristically defined graphs; rich road segment features including speed limits; and simultaneous information about traffic volume and speed. By capturing the complex road structure and traffic conditions of two major cities, we provide the community with the data needed for the development of holistic traffic forecasting systems and rigorous evaluation of corresponding models. Second, our empirical analysis reveals scalability issues in existing neural traffic forecasting models when applied to such large-scale traffic networks. To address these issues, we propose an efficient GNN-based approach that achieves superior scalability and forecasting performance.

**Future opportunities**    The proposed traffic datasets open several interesting avenues for future research. The first direction is the development of efficient traffic forecasting methods. While our proposed model shows decent performance, there is a continuous need to develop even better GNN architectures or alternative deep learning models that can efficiently process large urban road networks without sacrificing forecasting quality. Moreover, the presence of real connectivity structure and rich node features enables the development of models that can effectively exploit such information. Since our benchmark contains two different cities, it can be used to investigate how well models trained on one city (e.g., city-traffic-M) can generalize to another (e.g., city-traffic-L), which is a critical step towards universally applicable solutions for traffic forecasting. Our datasets provide a good starting point for such studies. Moreover, each of our datasets contains two dynamic variables — traffic volume and speed, which can be used to investigate the performance of forecasting models in multitask settings. The detailed forecasts enabled by these datasets could be directly integrated into adaptive traffic signal control systems, dynamic routing algorithms for logistics and navigation, and long-term urban infrastructure planning tools.

REPRODUCIBILITY STATEMENT

We provide all the details necessary to reproduce our experiments. Our datasets are shared via this private Kaggle link, and our code, including training and evaluation scripts, can be accessed in our anonymous GitHub repository. In Appendix D, we provide a detailed description of the experimental setup including hardware and software configurations.

For compliance with the double-blind review policy, we have shifted the original coordinates in the dataset. This transformation does not affect reproducibility since spatial features are standardized with a linear scaler, and thus the distribution of features remains unchanged. The only exception is Figure 2, which would appear distorted if plotted with shifted coordinates; for this figure we used the original coordinates to reconstruct the actual road topology via a standard projection of the coordinates from the Earth's sphere onto the plane using the haversine formula. We will release the original coordinates together with the public release of the datasets after the review process.

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

# A  DATASET DETAILS

The data used in our benchmark is collected from a widely-used online map and navigation service that estimates traffic congestion and travel time using high-resolution GPS signals transmitted by vehicles. To select the road segments, we take the central geographic point within each city, consider a circular area of a 15 kilometer radius, and include all road segments located within this area to the dataset. The obtained set of road segments includes the city itself and may also cover some nearby roads.

The traffic volume is estimated based on the number of vehicles that traverse each road segment during a specific timestamp interval, as inferred from aggregated GPS traces. It is important to note that the number of traverses represents an estimate rather than the actual traffic flow, as it is derived solely from vehicles equipped with GPS. Consequently, the reported values systematically underestimate the true traffic volume, but represent the dynamic of the traffic volume well. The speed estimation is also derived from these GPS signals, using a proprietary internal algorithm developed by the service provider.

Some characteristics of our datasets are reported in Table 5.

Table 5: Characteristics of new `city-traffic` datasets.

|  | city-traffic-M | city-traffic-L |
|---|---|---|
| # nodes | 53,530 | 94,009 |
| # edges | 121,236 | 164,424 |
| is directed | ✓ | ✓ |
| # timestamps | 35,449 | 35,449 |
| # train timestamps | 26,208 | 26,208 |
| # validation timestamps | 4,032 | 4,032 |
| # test timestamps | 5,209 | 5,209 |
| train start | Jul  1st 2024 00:00 | Jul  1st 2024 00:00 |
| validation start | Sep 30th 2024 00:00 | Sep 30th 2024 00:00 |
| test start | Oct 14th 2024 00:00 | Oct 14th 2024 00:00 |
| test end | Nov  1st 2024 02:00 | Nov  1st 2024 02:00 |
| avg. in-degree / avg. out-degree | 2.264 | 1.749 |
| avg. node degree (undirected) | 3.652 | 2.970 |
| Gini coefficient of degree distribution | 0.9 | 0.9 |

Each node in the dataset represents an individual road segment and has a set of 26 attributes, including categorical and binary indicators of road type, accessibility, and structural properties. The full list of feature names is the following:

- `category` — functional category of the road segment (e.g., major arterial, residential, service);
- `edge_type` — encodes the type of connection between the road segments;
- `speed_mode` — type of speed regulation pattern allowed on the segment (e.g., high-speed corridor, restricted-speed street);
- `speed_limit` — the maximum legal speed limit on the segment;
- `region_id` — identifier of the administrative or city district containing the segment;
- `can_bind_to_reverse_edge` — indicates whether the segment allows binding to a reverse-direction edge;
- `dismount_bike` — indicates if cyclists are required to dismount on the segment;
- `has_masstransit_lane` — indicates if the segment has a dedicated lane for public or mass transit;
- `ends_with_crosswalk` — indicates if the segment ends with a pedestrian crosswalk;

- `ends_with_toll_post` — indicates if the segment ends with a toll post;
- `is_in_poor_condition` — indicates whether the road surface is in poor condition;
- `is_paved` — indicates whether the segment is paved;
- `is_restricted_for_trucks` — indicates whether the segment is restricted for trucks;
- `is_toll` — indicates whether the segment is a toll road;
- `access_[0...5]`[1] — boolean masks for road accessibility by different undisclosed transport modes (exact mapping to vehicle types will be released by the provider);
- `length` — length of the road segment (in meters);
- `num_segments` — number of consecutive sub-segments composing the road segment;
- `x_coordinate_start` — latitude of the segment's start point;
- `y_coordinate_start` — longitude of the segment's start point;
- `x_coordinate_end` — latitude of the segment's end point;
- `y_coordinate_end` — longitude of the segment's end point.

Note that we apply ordinal encoding to the `speed_limit` feature. Thus, we provide the correspondence of particular feature values and their ordinal codes:

- $\text{NaN} \rightarrow 0$;
- $5\,km/h \rightarrow 1$;
- $20\,km/h \rightarrow 2$;
- $30\,km/h \rightarrow 3$;
- $40\,km/h \rightarrow 4$;
- $50\,km/h \rightarrow 5$;
- $60\,km/h \rightarrow 6$;
- $70\,km/h \rightarrow 7$;
- $80\,km/h \rightarrow 8$;
- $90\,km/h \rightarrow 9$;
- $100\,km/h \rightarrow 10$;
- $110\,km/h \rightarrow 11$;

---

[1]There is a separate feature for each of 6 masks.

## B DIFFERENCES BETWEEN CITY-TRAFFIC-M AND CITY-TRAFFIC-L

While both datasets follow the same construction methodology, there are several notable differences between city-traffic-M and city-traffic-L that make them complementary benchmarks.

In terms of scale, city-traffic-M contains 53,530 road segments and 121,236 directed edges, while city-traffic-L is almost twice as large, with 94,009 segments and 164,424 edges. The higher spatial resolution of city-traffic-L poses a particular challenge for the scalability of spatiotemporal models, as the number of graph nodes directly determines memory and runtime costs.

In terms of topological properties, the two cities also vary significantly and have a different urban structure. city-traffic-L features a complex structure shaped by a large river crossing the metropolitan area, which has led to the development of multiple islands connected by bridges. This creates bottlenecks and high-traffic corridors that models must capture. By contrast, city-traffic-M lacks such a riverine structure; its road network is more uniform, with a grid-like arrangement and wide avenues even in the central districts. Average node degree of a road network also differs between the datasets: city-traffic-M has an average undirected degree of 3.65, while city-traffic-L's average is 2.97. This reflects the higher density and branching structure of the smaller city versus the sparser but more geographically constrained connectivity of the larger one.

While the average traffic speed values are comparable between the two datasets, the same statistic for traffic volume differs significantly: city-traffic-L records substantially higher overall volume, reflecting its larger size. The weekly dynamics, shown in Figure 3, indicate more pronounced rush-hour congestion patterns in city-traffic-L.

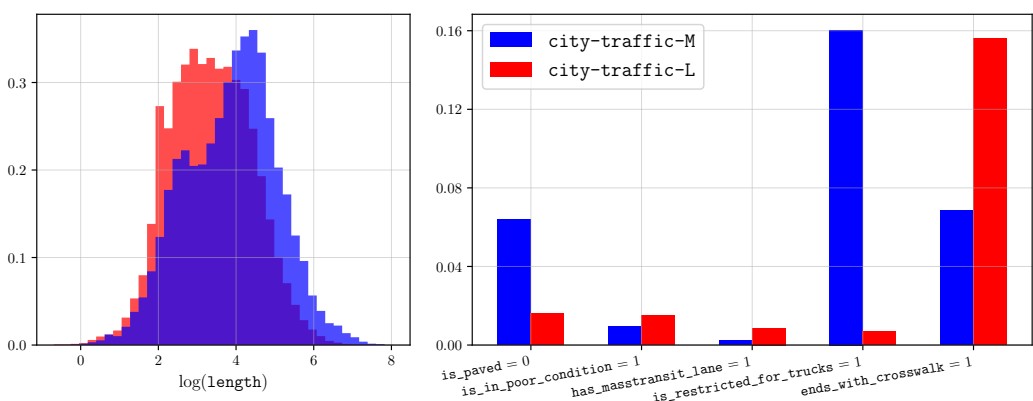

Figure 5: The distribution of some spatial features in the proposed datasets.

Both datasets provide the same 26 static attributes per segment. However, their distribution is different for the two proposed datasets. As Figure 5 shows, city-traffic-L has a greater fraction of paved roads, and there are also notably more roads with crosswalks at their endpoints. On the other hand, city-traffic-M has longer continuous road segments on average, and the fraction of roads restricted for trucks is much greater.

Taken together, the two datasets provide complementary perspectives: city-traffic-M highlights fine-grained dynamics in a compact road network, while city-traffic-L captures large-scale, heterogeneous urban traffic with more complex network structure. This difference is essential for developing models that generalize across diverse city types, rather than overfitting to one particular topology or traffic regime.

## C    RELATION BETWEEN SPATIAL ROAD FEATURES AND ROAD TRAFFIC

In this section, we provide several figures with the weekly dynamics of target variables for different road subsets depending on their static attributes and discuss how various spatial road features can affect the traffic volume and speed.

On Figure 6, we show the dynamics of target variables across the roads with a specific value of the `speed_limit` feature (in our case, we use the subset with $speed\_limit = 90\,km/h$ for both datasets). It can be seen that, on the roads with different speed limits, both traffic volume and traffic speed can vary significantly, as particular speed limit values can impose a notable restriction on the permitted traffic speed.

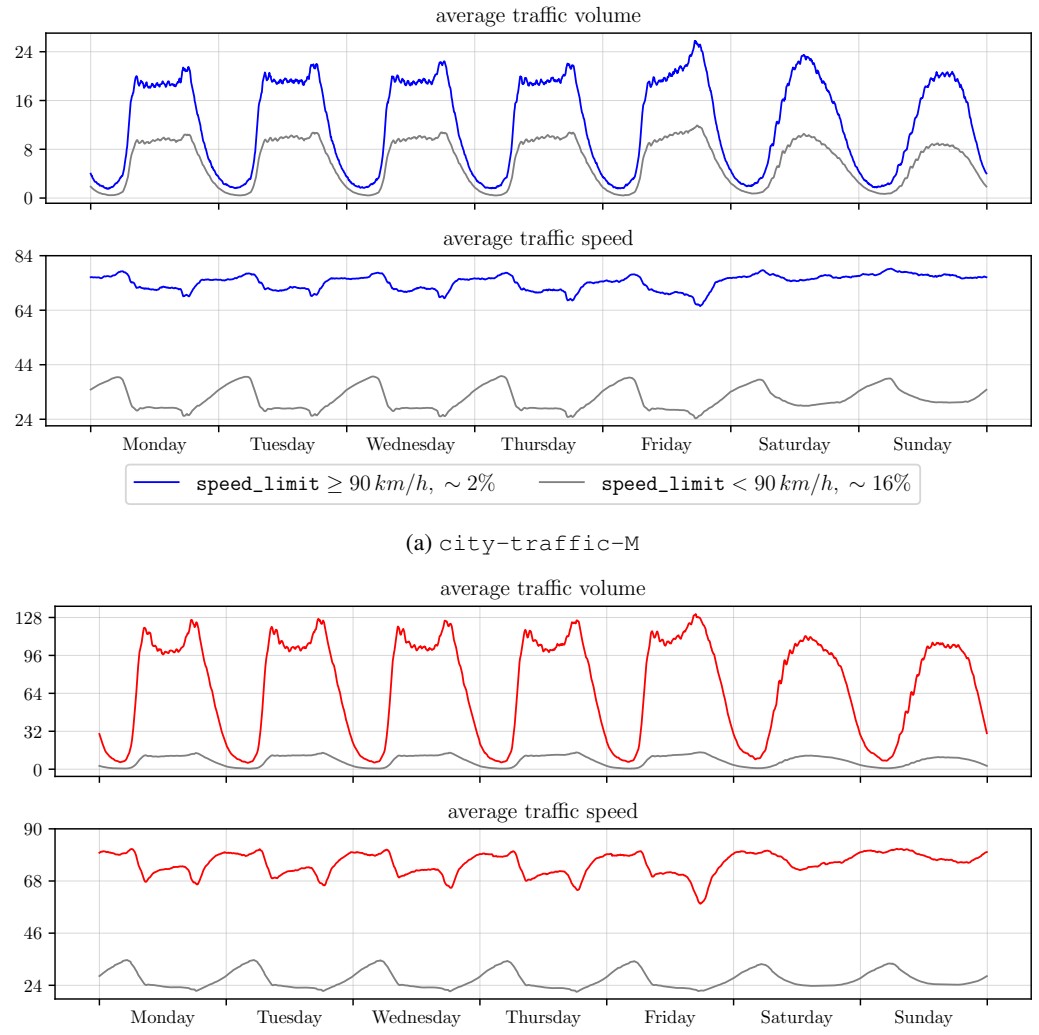

(a) `city-traffic-M`

(b) `city-traffic-L`

Figure 6: The weekly dynamics of target variables averaged across different road subsets depending on if they have $speed\_limit = 90\,km/h$. The percentage in the legend denotes the fraction of nodes in the corresponding category.

The next Figure 7 presents the target dynamics for the road subsets with different values of the `ends_with_crosswalk` feature. When moving on the roads that end with crosswalks, drivers have to slow down their vehicle in order to let pedestrians pass, which significantly affects the average traffic speed registered on such roads and makes it much lower on average.

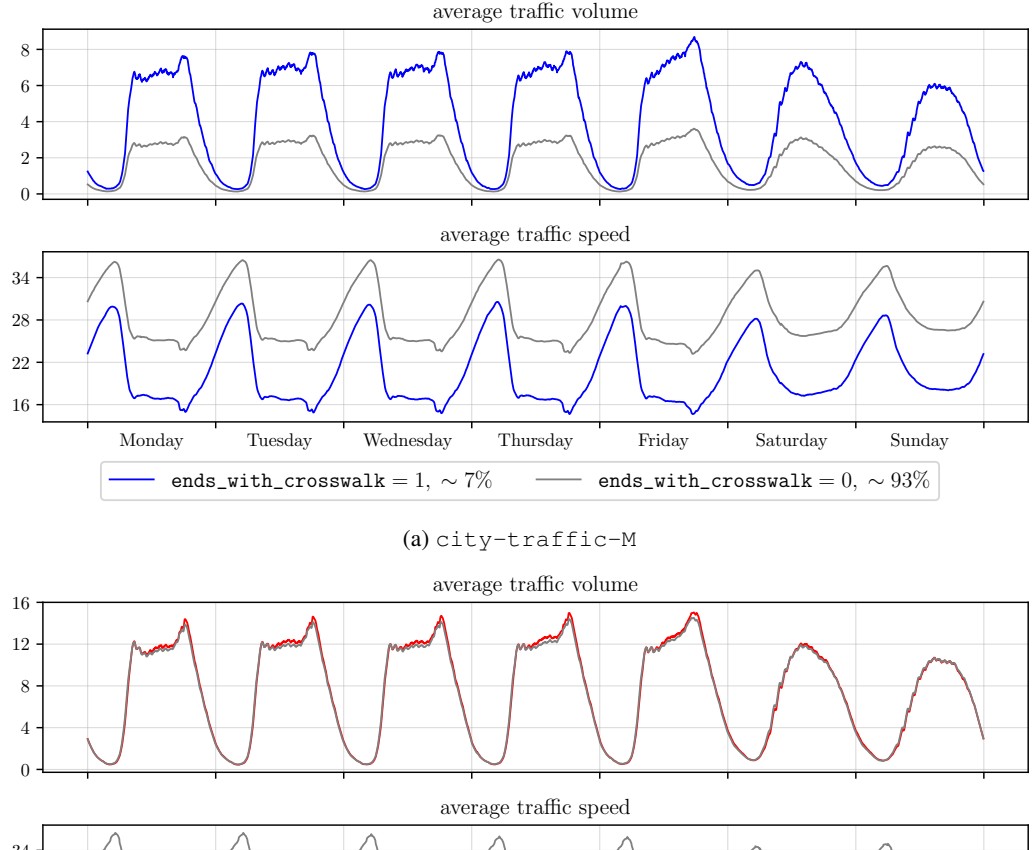

(a) `city-traffic-M`

(b) `city-traffic-L`

Figure 7: The weekly dynamics of target variables averaged across different road subsets depending on the value of `ends_with_crosswalk`. The percentage in the legend denotes the fraction of nodes in the corresponding category.

On Figure 8, we show the dynamics of targets variables for the subsets of roads that have different values of the `is_in_poor_condition` feature. If a road is in poor condition, drivers have to move on it more carefully and keep speed low in order to avoid any accidents. At the same time, there are not so many such roads in both cities, so traffic volume on the roads with normal condition is much higher on average.

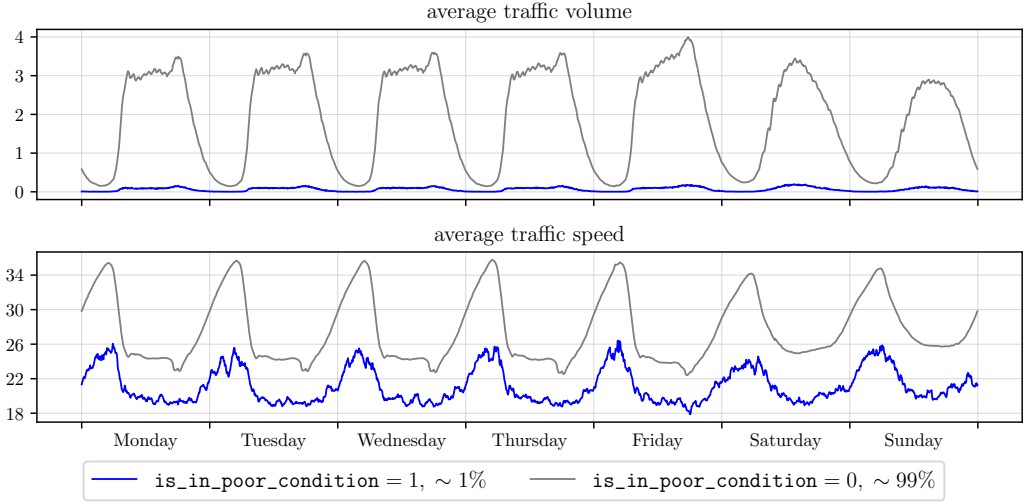

(a) `city-traffic-M`

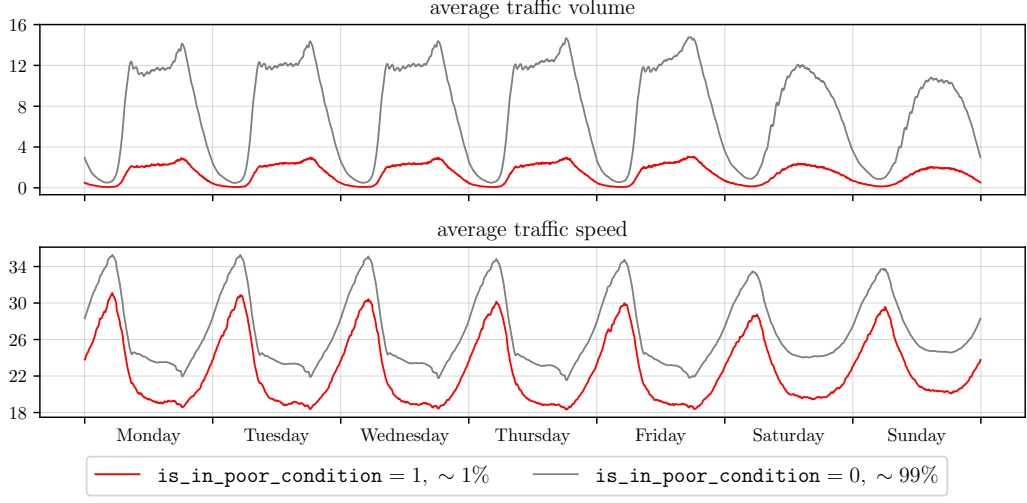

(b) `city-traffic-L`

Figure 8: The weekly dynamics of target variables averaged across different road subsets depending on the value of `is_in_poor_condition`. The percentage in the legend denotes the fraction of nodes in the corresponding category.

Figure 9 presents the target dynamics for the roads with different value of the `is_paved` feature. The movement on paved roads is more convenient and fast, which leads to higher traffic speed on average. Also, since pavement is a standard in road construction nowadays, the majority of roads in both cities have necessary surface, and most traffic volume is distributed exactly over paved roads.

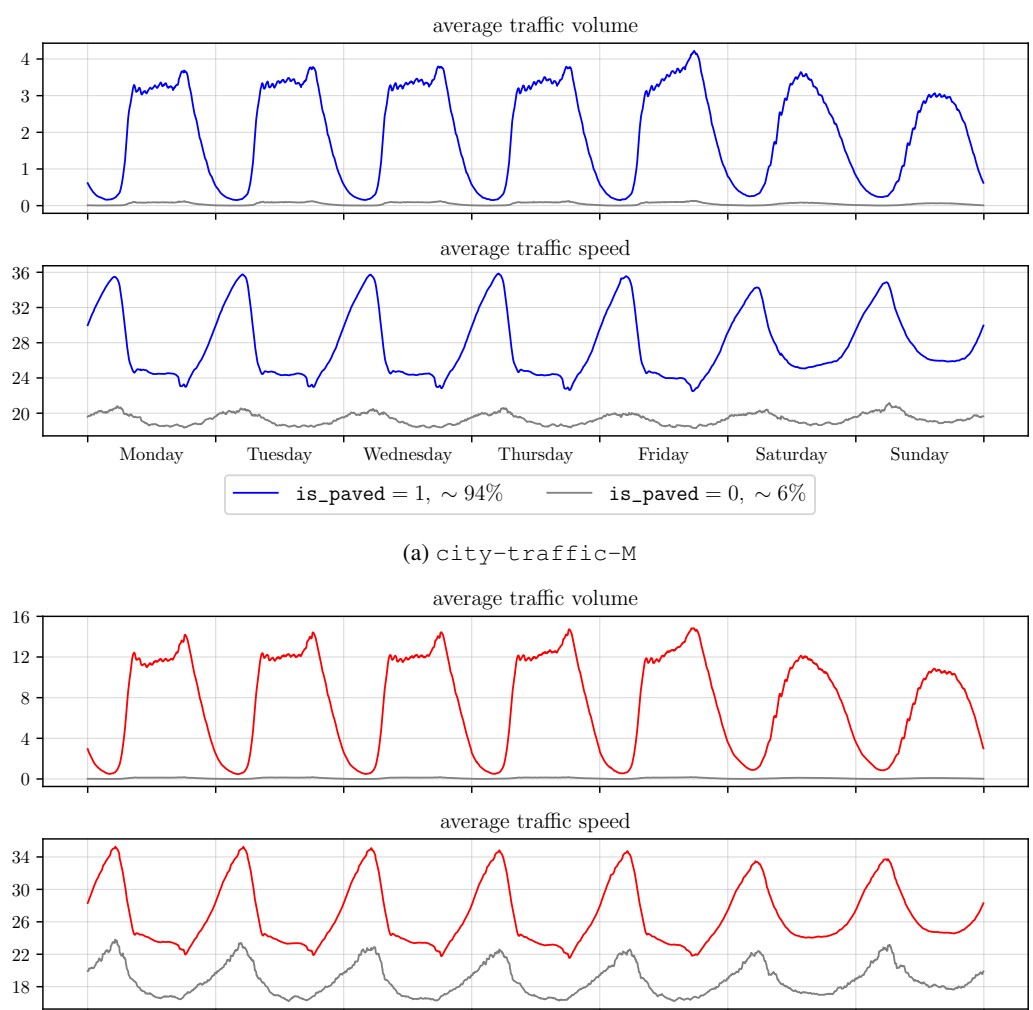

Figure 9: The weekly dynamics of target variables averaged across different road subsets depending on the value of `is_paved`. The percentage in the legend denotes the fraction of nodes in the corresponding category.

On Figure 10, we show the dynamics of targets variables across the roads with different values of the `length` feature. It is natural that on longer roads, drivers can afford moving on higher speed, in contrast to short roads that can connect different crossroads and crosswalks and may require to constantly slow down the vehicle. Moreover, since longer roads cover greater distance and typically connect locations with different logistic purpose in the larger city of `city-traffic-L`, they tend to carry more traffic volume.

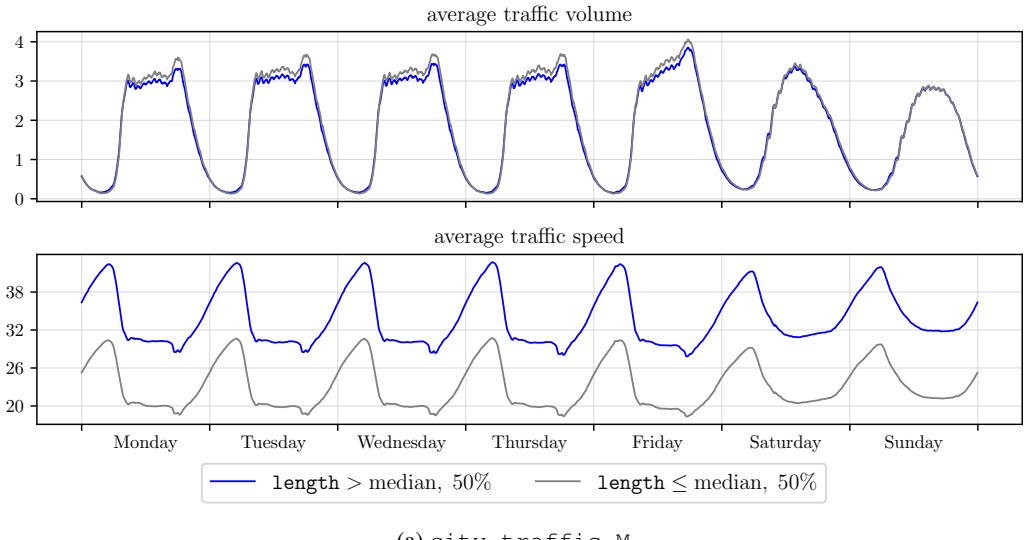

(a) `city-traffic-M`

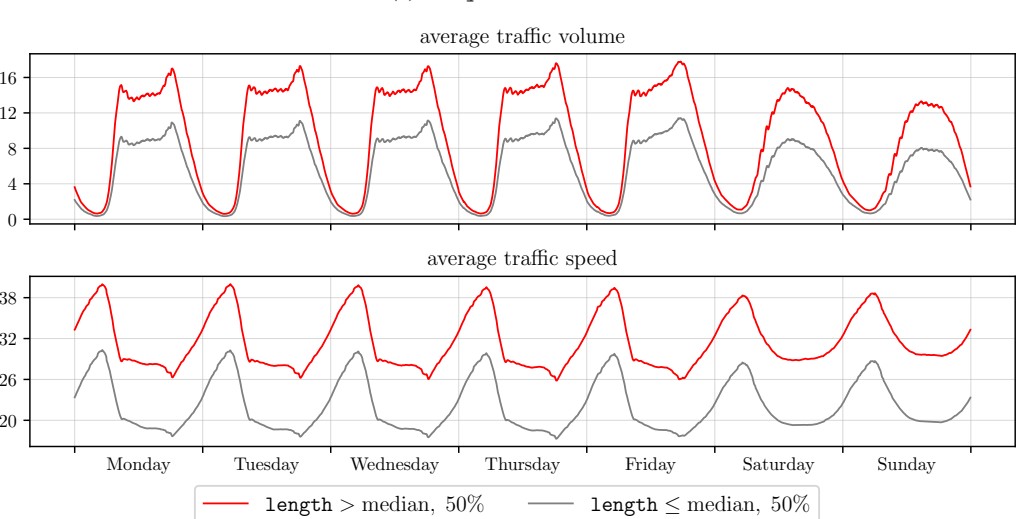

(b) `city-traffic-L`

Figure 10: The weekly dynamics of target variables averaged across different road subsets depending on the value of `length`. The percentage in the legend denotes the fraction of nodes in the corresponding category.

Figure 11 shows the target dynamics for the roads belonging to the central part of city (in our case, we decide to choose 25% of the roads) and to its periphery. In the city center, the structure of road network can be more complex and require more maneuvers to pass through it, so the average traffic speed on the central roads appears lower than on the peripheral ones. Further, since the city center in the smaller city of `city-traffic-M` has a more developed and diverse infrastructure that serves various needs of city residents, there is naturally more traffic volume.

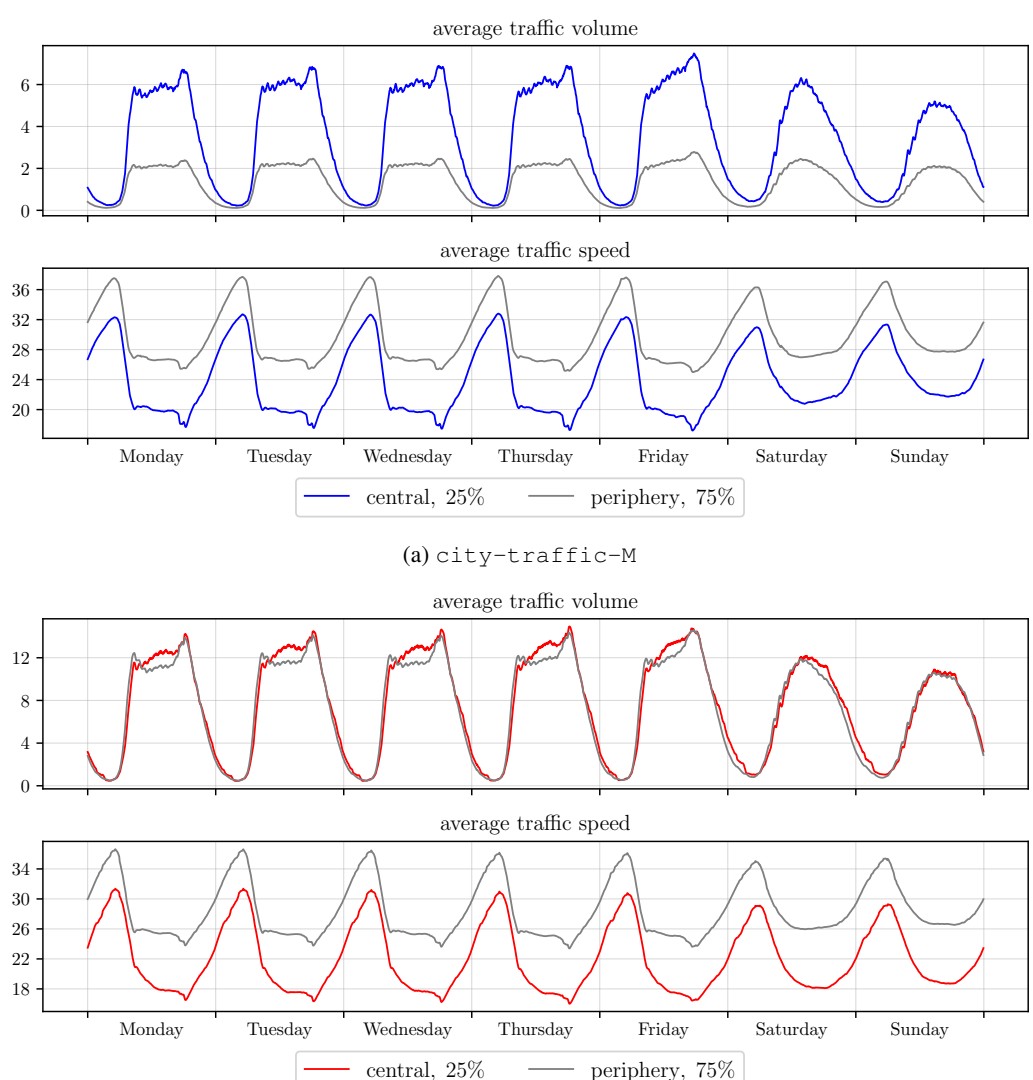

(a) `city-traffic-M`

(b) `city-traffic-L`

Figure 11: The weekly dynamics of target variables averaged across different road subsets depending on whether they are located at the city center.

The presented figures show that our proposed datasets contain important spatial information about road networks that has strong connection with the traffic speed and volume and thus is necessary to be used for precise traffic forecasting.

# D EXPERIMENTAL SETUP

In all our experiments, we train all models to predict next 12 timestamps of the temporal component.

We use learnable node embeddings for road segments in addition to their static features. We also use additional temporal calendar features such as day of the week and timestamp (hour and minute) during the day. We encode these features both with one-hot encoding and with periodic trigonometric functions.

For all models, we conducted hyperparameter search for the number of model blocks (layers) from 1 to 4 and over the hidden dimension from 32 to 512 (however, for less scalable models, larger values often led to out-of-memory issues).

All models are trained using the AdamW optimizer with a fixed learning rate of 0.0003. To ensure comparability across experiments, we fix the effective batch size to 30 across all datasets and adjust gradient accumulation steps as needed. Training is performed for 5 epochs, and each configuration is repeated three times. We report the mean and standard deviation of validation and test Mean Absolute Error (MAE), measured on designated subsets of timestamps.

All experiments are constrained to a single A100 GPU with 80GB of VRAM and 120GB of system RAM. For failed combinations, we try to decrease the number of model parameters — after several attempts, if the configuration still fails, we exclude it from the comparison. All experiments use a full-batch training mode without neighbor sampling. This choice is motivated by the need for consistent and fair comparison between models, particularly because neighbor sampling introduces stochasticity that can disproportionately affect certain architectures and complicate evaluation. Moreover, given the scale of our datasets and the memory available on a single GPU, full-batch training remains feasible and provides deterministic gradient computations that improve stability and reproducibility.

We use `dgl==2.4.0+cu124` and `torch==2.4.0+cu124` for our experiments.

# E DETAILS OF THE PROPOSED EFFICIENT SPATIOTEMPORAL FORECASTING NEURAL NETWORK ARCHITECTURE

In this section, we describe in more detail the efficient neural network architecture used for GNN-`Mean` and GNN-`TrfAttn`.

Let $x_i \in \mathbb{R}^F$ be the input feature vector for node $i$, which contains static node features (road segment coordinates, road segment speed limit, etc; see Appendix A for more details), learnable node embeddings, temporal calendar features (day of the week, etc.), and past values of traffic speed and volume in the lookback window (all these features are concatenated in a single vector of dimension $F$). First, our architecture transforms this input feature vector into a single hidden representation $h_i^0 \in \mathbb{R}^H$ (where $H$ is the hidden dimension) with a linear layer, followed by dropout (Srivastava et al., 2014) and a GELU activation function (Hendrycks & Gimpel, 2016):

$$h_i^0 = \text{GELU}(\text{Dropout}(\text{Linear}(x_i))).$$

Then, this hidden representation is iteratively transformed by $L$ GNN blocks (where $L$ is the number of blocks/layers):

$$h_i^{l+1} = \text{GNNBlock}(h_i^l).$$

Finally, the hidden representation obtained from the last GNN block is normalized with a layer normalization (Ba et al., 2016) and transformed to predictions $\widehat{y}_i$ with a linear layer:

$$\widehat{y}_i = \text{Linear}(\text{LayerNorm}(h_i^L)).$$

Our GNN block is based on the architecture from Platonov et al. (2023); Bazhenov et al. (2025), and, besides a neighborhood aggregation operation, includes a residual connection (He et al., 2016), a layer normalization, and a 2-layer MLP. We also concatenate the result of neighborhood aggregation with the representation of the aggregating node, similar to GraphSAGE (Hamilton et al., 2017). Let $N(i)$ be the set of 1-hop neighbors of node $i$ in the graph. Then, a single GNN block can be described as follows:

$$h_i^{\text{norm}} = \text{LayerNorm}(h_i^{\text{input}}),$$

$$h_i^{\text{aggr}} = \text{Concatenate}(h_i^{\text{norm}}, \text{Aggregate}(h_i^{\text{norm}}, \{h_j^{\text{norm}} \ \forall j : j \in N(i)\})),$$

$$h_i^{\text{MLP}} = \text{MLP}(h_i^{\text{aggr}}),$$

$$h_i^{\text{output}} = h_i^{\text{input}} + h_i^{\text{MLP}}.$$

The neighborhood aggregation operation is what is different between the two models used in our work. For GNN-`Mean`, it simply takes the mean of the neighbor representations:

$$\text{Aggregate}_{\text{Mean}}(h_i^{\text{norm}}, \{h_j^{\text{norm}} \ \forall j : j \in N(i)\}) = \text{Mean}(\{h_j^{\text{norm}} \ \forall j : j \in N(i)\}).$$

For GNN-`TrfAttn`, it is a multihead scaled dot product attention (Vaswani, 2017) with queries obtained from the aggregating node $i$ and keys and values obtained from its neighbors:

$$\text{Aggregate}_{\text{TrfAttn}}(h_i^{\text{norm}}, \{h_j^{\text{norm}} \ \forall j : j \in N(i)\})$$
$$= \text{MultiheadAttention}(\text{source} = h_i^{\text{norm}}, \text{targets} = \{h_j^{\text{norm}} \ \forall j : j \in N(i)\}).$$

## F    LIMITATIONS

While our benchmark provides novel and valuable data for fine-grained urban traffic forecasting, it has certain limitations. First, we acknowledge that cities in different countries may exhibit different traffic patterns. Consequently, the conclusions drawn from the two cities included in our benchmark may not be directly generalizable to urban environments with substantially different traffic dynamics. Additionally, as our benchmark spans only four months of data, it may not facilitate the evaluation of models designed to capture long-term annual trends. However, we contend that for urban traffic, which is characterized by rapidly changing conditions, the ability to capture local trends, such as recent traffic conditions on a specific road segment and its adjacent segments, is often more critical.

## G    LLM USAGE

LLMs have been used to aid with polishing the writing of this manuscript. LLMs were not used to generate research ideas, results, figures, or numerical values reported in tables. All experiments, datasets, and analyses are fully conducted by the authors.

