# OpenReview forum: "Fine-Grained Urban Traffic Forecasting on Metropolis-Scale Road Networks"
_ICLR.cc/2026/Conference — Submitted to ICLR 2026_

### Official Review · Reviewer_YjaT · 2025-10-30

**Soundness:** 2
**Presentation:** 2
**Contribution:** 2
**Rating:** 2
**Confidence:** 5

**Summary:**

This paper introduces two large-scale, fine-grained urban traffic forecasting datasets (city-traffic-M with 53k road segments and city-traffic-L with 94k segments) derived from GPS traces in two major (anonymized) cities. These datasets provide real road graph connectivity, rich static road features, and dual targets (5-min granularity traffic speed and volume) over ~4 months. They starkly contrast prior benchmarks (e.g., METR-LA: 207 sparse highway sensors; LargeST: 8.6k), which lack urban density, true graphs, and volume data. The authors benchmark popular spatiotemporal GNNs (DCRNN, GRUGCN, STGCN, GWN), showing most fail to scale (memory/time blow up on 80GB GPU). They propose a scalable alternative: per-node linear flattening of the lookback window into a single embedding, followed by a lightweight GNN (mean or transformer-attention aggregation, with skips/LN/MLPs). This "time-then-graph" design has O(nd) memory (n=nodes, d=dim), enabling longer horizons cheaply. It outperforms baselines in accuracy while training faster.

**Strengths:**

1- Large-scale dataset: The released dataset is openly available, metropolis-scale urban traffic datasets, 10-500x larger than priors (e.g., METR-LA: 207; LargeST: 8.6k) with true directed road graphs (adjacency via traffic rules), 26 rich static features (length, speed limits, surface quality, endpoints, transit lanes), and dual 5-min GPS targets (speed + volume, Jul-Nov 2024; realistic 5-25% speed missingness). Urban-dense (Figs. 1-2) vs. priors' sparse highways; first speed+volume (Table 1).

2- Extensive and Reproducible Experiments: Experiments compare against major baselines (DCRNN, STGCN, GRUGCN, GWN) under strict GPU constraints. Ablation on lookback window length, scalability tests, and runtime benchmarks are provided. Clear documentation of training environment (single A100 GPU, full-batch training) and public code/data links.

**Weaknesses:**

1 - Experimental Limitations: i) Metrics are narrow: Only MAE, no MAPE/RMSE/MSE (std for traffic); ii) Horizons missing: Fixed prediction length (1-step?), no 15/30/60min (critical for routing); iii) Baselines dated: Miss recent (Chronos/TimesFM-GNN; MTGNN; STNorm); no multi-task (joint speed+vol); no extrinsics (events/holidays). No inference latency; CPU/FLOPs; zero-shot roads.

2- Narrow Evaluation Scope: All experiments are conducted on the two proposed datasets only. The model is not evaluated on existing public datasets (e.g., METR-LA, PeMS-BAY), which limits direct performance comparison and external validity. No cross-city transfer or domain generalization experiments are shown, despite mentioning this as a future direction.

3 - Limited Long-Term Dynamics: The dataset covers only four months, restricting the evaluation of seasonal and long-term forecasting. The authors acknowledge this but do not discuss how short data duration affects model generalization to yearly cycles.

4 - Research Scope: most importantly, in my opinion, providing the research community a large time series traffic dataset is a good contribution; nevertheless, I wonder if this research focus is suitable for the ICLR conference, which mainly focuses on learning representation. Please emphasize the main contribution of this work for better clarification.

**Questions:**

Please see the comments in Weaknesses Section.

---

> ### Author Response · Authors · 2025-11-21
> **Authors' response pt. 1/3**
>
> Thank you for your detailed feedback! We address your questions and concerns below.
>
> > Metrics are narrow: Only MAE, no MAPE/RMSE/MSE (std for traffic);
>
> Thank you for this suggestion. We additionally provide MSE, RMSE, MAPE, and $R^2$ values. Originally, we focused on MAE because it is standard in spatiotemporal forecasting and easy to interpret in the original units while being less sensitive to rare extreme outliers.
>
> We observe that across most of the reported metrics and datasets, our proposed GNN-TrfAttn consistently outperforms both the linear model and spatiotemporal baselines. For example, on `city-traffic-L-speed`, GNN-TrfAttn improves RMSE from 4.20 (DCRNN) and 4.96 (Linear) to 4.08. During the discussion period, we also added the remaining baselines (STGCN and GWN), which took more time due to computational demanding architecture.
>
> | Metric | Model | city-traffic-L-speed | city-traffic-L-volume | city-traffic-M-speed | city-traffic-M-volume |
> |------|-----|---------|------------|---------|----|
> | MAPE | DCRNN | 46.3766±0.7652 | 45.8673±0.2413 | 20.0628±0.1169|53.9618±0.3088|
> | MAPE | GRUGCN | 47.0907±0.1525 | 46.5107±0.1026 | 21.2006±0.1117 | 53.8142±0.5566|
> | MAPE | STGCN |MLE |MLE | 22.5196±0.1756 | 53.6175±0.208 |
> | MAPE | GWN | 48.4992±1.6725 | 49.4875±0.5876 | 25.3965±0.5872 | 55.1374±0.2351 |
> | MAPE | LinearModel | 50.1822±0.2093 | 50.0899±0.2053 | 21.8939±0.0494 | 52.9170±0.0172 |
> | MAPE | GNN-Mean | 47.6612±0.827 | 47.1111±0.6591 | 19.2586±0.4517 | 53.3307±0.6243 |
> | MAPE | GNN-TrfAttn | 46.0017±1.0215 | 47.1618±0.3249 | 19.1637±0.3551 | 52.6833±0.1886 |
> | MSE | DCRNN | 17.622±0.544|17.7813±0.6246|26.5022±0.0408|2.9401±0.041|
> | MSE | GRUGCN | 27.1225±3.6369 | 28.4905±4.0186 | 27.9145±0.2589 | 3.2693±0.0975 |
> | MSE | STGCN |MLE |MLE | 30.8621±0.3538|3.0126±0.0335|
> | MSE | GWN | 25.7573±2.351|27.1942±4.3626|36.1415±0.3581|3.661±0.0819|
> | MSE | LinearModel | 24.5953±0.0273 | 24.5954±0.0281 | 31.2399±0.0461 | 3.5939±0.0014 |
> | MSE | GNN-Mean | 17.3032±0.3436 | 17.1190±0.1620 | 25.1117±0.1970 | 2.8578±0.0125 |
> | MSE | GNN-TrfAttn | 16.6551±0.1501 | 16.8240±0.0955 | 24.8780±0.2399 | 2.7884±0.0233 |
> | $R^2$ | DCRNN | 0.9496±0.0016 | 0.9491±0.0018 | 0.8893±0.0002 | 0.9000±0.0014 |
> | $R^2$ | GRUGCN | 0.9224±0.0104 | 0.9185±0.0115 | 0.8834±0.0011 | 0.8888±0.0033 |
> | $R^2$ | STGCN |MLE |MLE | 0.8711±0.0015| 0.8976±0.0011|
> | $R^2$ | GWN | 0.9263±0.0067| 0.9222±0.0125|0.849±0.0015|0.8755±0.0028|
> | $R^2$ | LinearModel | 0.9296±0.0001 | 0.9296±0.0001 | 0.8695±0.0002 | 0.8778±0.0000 |
> | $R^2$ | GNN-Mean | 0.9505±0.0010 | 0.9510±0.0005 | 0.8951±0.0008 | 0.9028±0.0004 |
> | $R^2$ | GNN-TrfAttn | 0.9523±0.0004 | 0.9519±0.0003 | 0.8961±0.0010 | 0.9052±0.0008 |
> |RMSE|DCRNN|4.1974±0.0645|4.2162±0.0741|5.148±0.004|1.7146±0.0119|
> | RMSE | GRUGCN | 5.1979±0.3425 | 5.3265±0.3772 | 5.2834±0.0245 | 1.8079±0.0269 |
> | RMSE | STGCN |MLE |MLE | 5.5553±0.0318|1.7357±0.0096|
> | RMSE | GWN | 5.0706±0.2284|5.2005±0.4089|6.0117±0.0298|1.9133±0.0214|
> | RMSE | LinearModel | 4.9594±0.0028 | 4.9594±0.0028 | 5.5893±0.0041 | 1.8958±0.0004 |
> | RMSE | GNN-Mean | 4.1595±0.0414 | 4.1375±0.0196 | 5.0111±0.0197 | 1.6905±0.0037 |
> | RMSE | GNN-TrfAttn | 4.0810±0.0184 | 4.1017±0.0116 | 4.9877±0.0240 | 1.6698±0.0070 |
>
> _Performance of different methods on `city-traffic` datasets._ MLE indicates CUDA Memory Limit Error
>
>
> > Horizons missing: Fixed prediction length (1-step?), no 15/30/60min (critical for routing)
>
> We would like to clarify that our default setting reported in all the tables is forecasting for 12 steps (1 hour) ahead and averaging metrics over these 12 steps. This was mentioned in Appendix D but without specifying particular timestamps used. We now clarified that, thank you for noticing!

---

> ### Author Response · Authors · 2025-11-21
> **Authors' response pt. 2/3**
>
> > Baselines dated: Miss recent (Chronos/TimesFM-GNN; MTGNN; STNorm)
>
> Please note that these are models for classic time series forecasting, not graph-aware spatiotemporal forecasting on which we focus in our work. Adapting foundation time-series models such as Chronos and TimesFM to large road graphs would require designing an additional spatial component and substantial engineering to make them scale on 50K-100K-node graphs, which is beyond the scope of our dataset-focused work. Our benchmark is, however, fully compatible with such future hybrids (e.g., Chronos-style temporal backbone + GNN over our road graphs), and one of our goals is precisely to enable the community to explore such combinations.
>
> However, we have also managed to run an experiment with a recent model LSTTN [5] on our medium-scale datasets `city-traffic-M`. Because of the limited computational resources, most configurations of its architecture could not meet the specified memory constraints, and we had to modify the original source code to address some technical issues. After training for the required number of epochs, it achieved 10.670 MAE points on `city-traffic-M-speed` and 2.718 MAE points on `city-traffic-M-volume`, yielding the results only comparable to the most simple baselines, such as constant prediction. This again suggests that previous sequential methods for traffic forecasting may not be ready for more complex and large-scale road networks, such as those introduced in our work, and require further adjustments to solve the problem of their efficiency.
>
>
> [1] LSTTN: A Long-Short Term Transformer-based spatiotemporal neural network for traffic flow forecasting (Knowledge-Based Systems 2024)
>
> > no multi-task (joint speed+vol)
>
> To the best of our knowledge, no prior work on traffic forecasting considers a multitask setup, so we leave the investigation of this setup for future work. Our primary goal in this paper is to introduce a realistic benchmark and evaluate representative single-task setups, following existing literature.
>
>
>
> > No inference latency; CPU/FLOPs
>
> All our experiments are run in `bf16` on a single NVIDIA A100 GPU with 80GB VRAM using Tensor Cores. Given the theoretical `bf16` throughput of approximately 624 TFLOPs/s on this hardware, we can upper-bound the total training compute for each model as:
>
> $$TFLOPS_{train} \le 624 \times T_{seconds} \le 2.24 \times 10^{6} \times T_{hours},$$
>
> where $T_\text{hours}$ is the training time reported in Table 4 for a given dataset and lookback. From Table 4, our GNN-Mean and GNN-TrfAttn models typically train in **1–3 hours** per configuration, whereas sequential baselines such as DCRNN, GWN, and especially STGCN require **tens to hundreds of hours** (STGCN even hits a 250-hour timeout on city-traffic-L with lookback 48). Since total `bf16` compute is proportional to training time on the same GPU, this implies roughly **one to two orders of magnitude fewer FLOPs** for our models compared to these baselines on our large graphs. We will add this formula and the corresponding `bf16` compute estimates (derived directly from the times in Table 4) to the appendix, alongside the wall-clock inference times.
>
> Below we report wall-clock inference latency for a full evaluation run on each dataset. We convert seconds to minutes rounded to 2 decimals. MLE means that we were unable to get any measurements of latency for `city-traffic-L` for STGCN due to its high computational demand.
>
> | Model| `city-traffic-L` latency (min) | `city-traffic-M` latency (min) |
> |-----|------|---------|
> | Linear Model | 4.08| 1.28|
> | DCRNN| 90.88| 63.00|
> | GRUGCN| 18.30| 63.00|
> | STGCN| MLE| 19.57|
> | GWN| 51.00| 25.00|
> | GNN-Mean| **4.28**| **3.05**|
> | GNN-TrfAttn| **5.83**| **3.00**|
>
> The table shows that sequential baselines are prohibitively slow at metropolis scale: DCRNN needs **90.88/63.00 min** and GWN **51.00/25.00 min**, while STGCN does not finish on `city-traffic-L` and still takes **19.57 min** on `city-traffic-M`. In contrast, our efficient models run in **~3–6 min** (GNN-Mean **4.28/3.05**, GNN-TrfAttn **5.83/3.00**), giving **an order-of-magnitude faster inference** while maintaining better accuracy.
>
>
> > zero-shot roads
>
> We believe such a setup will not be realistic, as road networks are generally rather static with no new roads suddenly appearing. However, we conducted an additional experiment in which we try to zero-shot transfer a model trained on one city to the other city, see our reply to your question about this below. This setup seems closer to realistic deployment, where a model trained for one metropolitan area might be reused for another.

---

> ### Author Response · Authors · 2025-11-21
> **Authors' response pt. 3/3**
>
> > All experiments are conducted on the two proposed datasets only. The model is not evaluated on existing public datasets (e.g., METR-LA, PeMS-BAY), which limits direct performance comparison and external validity.
>
> Our main goal is to introduce large scale realistic datasets that can stress-test model scalability. Our efficient spatiotemporal forecasting framework is designed specifically for large-scale settings. Prior datasets METR-LA and PEMS-BAY are extremely small (300-500 nodes) and thus do not require efficient models.
>
> > No cross-city transfer or domain generalization experiments are shown, despite mentioning this as a future direction.
>
> We provide some preliminary results in this direction below. Concretely, we train GNN-TrfAttn on `city-traffic-L-speed `and evaluate it zero-shot on `city-traffic-M-speed`. To make this setup meaningful, we exclude city-specific static features (absolute coordinates, `region_id`, and city-specific road categories), so that the model relies only on generic attributes that can, in principle, transfer across cities. The resulting metrics on `city-traffic-M-speed` are the following:
> - MSE: 55.9729
> - RMSE: 7.4815
> - MAE: 5.7171
> - MAPE: 42.8260
> - $R^2$: 0.7705
>
> For comparison, the same model evaluated in-distribution on `city-traffic-M-speed` in our main experiments achieves $\text{MSE} \approx 24.9$ and $R^2 \approx 0.90$. Thus, cross-city transfer roughly doubles the MSE and reduces $R^2$ by about 0.13, indicating that domain shift between the two cities is substantial and that cross-city generalization is a challenging open problem for the field.
>
> > Limited Long-Term Dynamics: The dataset covers only four months, restricting the evaluation of seasonal and long-term forecasting. The authors acknowledge this but do not discuss how short data duration affects model generalization to yearly cycles.
>
> First, we would like to note that prior most popular datasets METR-LA and PEMS-BAY also only cover 4 and 6 months respectively. More importantly, our datasets are obtained from a real-world traffic monitoring and forecasting system and closely follow its forecasting setting, and this system in fact also uses only a few months of prior data for forecasting, which turned out to be the best variant in internal evaluations and thus captures all the relevant traffic patterns in practice. Therefore, our datasets represent the setup of a real-world application. Finally, due to fine-grained spatial and temporal resolutions, our datasets are already very large and it is challenging to scale existing models to them: training some of these models takes multiple days. Increasing the covered time period will significantly exacerbate this issue.
>
> > Research Scope: most importantly, in my opinion, providing the research community a large time series traffic dataset is a good contribution; nevertheless, I wonder if this research focus is suitable for the ICLR conference, which mainly focuses on learning representation. Please emphasize the main contribution of this work for better clarification.
>
> Indeed, our work focuses first and foremost on two new large-scale and realistic datasets. Please note that the ICLR 2026 call for papers lists “Datasets and Benchmarks” as one of the relevant topics for the conference. This shows that our paper fits the scope of the conference, and we have appropriately chosen “Datasets and Benchmarks” as the primary area of our paper. Since empirical ML research is impossible without high-quality datasets, we believe dataset papers are of great importance for the research community. From a representation learning perspective, our datasets expose a learning setting that was previously inaccessible: metropolis-scale road graphs with real connectivity and rich attributes, on which many existing spatiotemporal GNNs fail to scale. Our empirical study thus provides concrete evidence about which graph/temporal representations remain viable at this scale and shows that a much simpler linear temporal encoding can outperform more complex sequential modules. We will clarify in the paper that the core contribution is a new benchmark for learning scalable spatiotemporal representations on realistic urban road networks, with the proposed model serving as an illustrative solution.

---

> ### Author Response · Authors · 2025-12-04
> **Author's response on additional separate metrics**
>
> Dear Reviewer, we additionally address your concerns regarding metrics on the specific timestamps
>
> For our setup of predicting 12 timestamps ahead, we also report the metrics computed on `city-traffic-L speed` for each timestamp separately for GNN-TrfAttn, GRUGCN and a naive baseline using the previous value. As can be seen, all metrics get worse with increasing prediction horizon, which is intuitive behaviour, as further timestamps are harder to predict based on the existing historical window.
>
> Nevertheless, our model achieves better metric on each timestamp:
>
> | Horizon  | GNN-TrfAttn MAE ↓    | GRUGCN MAE ↓        | Previous-val MAE ↓ |
> |-----|----|---|--|
> | MAE 5min | 1.8369 ± 0.0139      | 1.9821 ± 0.0265     | 2.3048|
> | MAE 10min| 1.8999 ± 0.0487      | 1.9853 ± 0.0266     | 2.3354|
> | MAE 15min| 1.9568 ± 0.0429      | 2.0270 ± 0.0206     | 2.4338|
> | MAE 20min| 1.9476 ± 0.0300      | 2.0489 ± 0.0210     | 2.4920|
> | MAE 25min| 1.9977 ± 0.0352      | 2.0709 ± 0.0282     | 2.6001|
> | MAE 30min| 1.9575 ± 0.0094      | 2.0804 ± 0.0294     | 2.6321|
> | MAE 35min| 1.9702 ± 0.0082      | 2.1068 ± 0.0248     | 2.7123|
> | MAE 40min| 2.0554 ± 0.0823      | 2.1264 ± 0.0308     | 2.7484|
> | MAE 45min| 2.0433 ± 0.0339      | 2.1549 ± 0.0304     | 2.8220|
> | MAE 50min| 2.0239 ± 0.0150      | 2.1726 ± 0.0327     | 2.8434|
> | MAE 55min| 2.0422 ± 0.0350      | 2.1981 ± 0.0301     | 2.9295|
> | MAE 60min| 2.0618 ± 0.0092      | 2.2195 ± 0.0398     | 2.9659|
>
> _MAE metric for each timestamp separately_
>
>
> | Horizon   | GNN-TrfAttn MAPE ↓      | GRUGCN MAPE ↓           | Previous-val MAPE ↓ |
> |-----------|-------------------------|--------------------------|---------------------|
> | MAPE 5min | 45.1070 ± 1.2298        | 45.2788 ± 0.1583         | 58.77|
> | MAPE 10min| 43.8280 ± 0.6109        | 45.6072 ± 0.3383         | 59.32|
> | MAPE 15min| 45.4001 ± 1.8850        | 45.6388 ± 0.3308         | 60.49|
> | MAPE 20min| 45.9658 ± 1.7036        | 46.4386 ± 0.4440         | 61.38|
> | MAPE 25min| 45.1443 ± 1.4250        | 46.5920 ± 0.3570         | 62.65|
> | MAPE 30min| 46.2271 ± 1.3202        | 46.6623 ± 0.2828         | 63.34|
> | MAPE 35min| 46.0574 ± 1.1184        | 46.7400 ± 0.4925         | 64.43|
> | MAPE 40min| 48.0980 ± 4.1764        | 47.0525 ± 0.3282         | 65.16|
> | MAPE 45min| 45.4231 ± 0.4355        | 47.4752 ± 0.5524         | 66.23|
> | MAPE 50min| 47.6885 ± 1.6175        | 47.7529 ± 0.3993         | 66.93|
> | MAPE 55min| 46.7075 ± 1.1588        | 48.0139 ± 0.6497         | 68.13|
> | MAPE 60min| 46.3728 ± 2.1926        | 48.3008 ± 0.6405         | 68.99|
> _MAPE metric for each timestamp separately_
>
>
> | Horizon    | GNN-TrfAttn RMSE ↓     | GRUGCN RMSE ↓           | Previous-val RMSE ↓ |
> |------------|------------------------|--------------------------|---------------------|
> | RMSE 5min  | 3.6593 ± 0.0174        | 5.0060 ± 0.3898          | 4.5296|
> | RMSE 10min | 3.7538 ± 0.0122        | 4.8430 ± 0.3768          | 4.6204|
> | RMSE 15min | 3.8676 ± 0.0078        | 4.9888 ± 0.3231          | 4.9056|
> | RMSE 20min | 3.9599 ± 0.0144        | 5.1334 ± 0.3357          | 5.0801|
> | RMSE 25min | 4.0400 ± 0.0158        | 5.1341 ± 0.3933          | 5.3968|
> | RMSE 30min | 4.0856 ± 0.0162        | 5.1128 ± 0.3796          | 5.4979|
> | RMSE 35min | 4.1379 ± 0.0059        | 5.1854 ± 0.3050          | 5.7382|
> | RMSE 40min | 4.2033 ± 0.0559        | 5.2804 ± 0.3654          | 5.8554|
> | RMSE 45min | 4.2169 ± 0.0206        | 5.3879 ± 0.3896          | 6.0785|
> | RMSE 50min | 4.2738 ± 0.0347        | 5.4237 ± 0.3437          | 6.1629|
> | RMSE 55min | 4.3317 ± 0.0244        | 5.4656 ± 0.3192          | 6.4219|
> | RMSE 60min | 4.3723 ± 0.0317        | 5.4493 ± 0.3841          | 6.5535|
> _RMSE metric for each timestamp separately_
>
>
> | Horizon  | GNN-TrfAttn $R^2$ ↑        | GRUGCN $R^2$ ↑             | Previous-val $R^2$ ↑ |
> |-|-|-|-|
> | $R^2$ 5min  | 0.9671 ± 0.0004         | 0.9279 ± 0.0115         | 0.9441           |
> | $R^2$ 10min | 0.9597 ± 0.0003         | 0.9325 ± 0.0107         | 0.9388           |
> | $R^2$ 15min | 0.9572 ± 0.0002         | 0.9283 ± 0.0094         | 0.9310           |
> | $R^2$ 20min | 0.9551 ± 0.0003         | 0.9243 ± 0.0100         | 0.9260           |
> | $R^2$ 25min | 0.9533 ± 0.0004         | 0.9242 ± 0.0116         | 0.9165           |
> | $R^2$ 30min | 0.9522 ± 0.0004         | 0.9248 ± 0.0114         | 0.9133           |
> | $R^2$ 35min | 0.9510 ± 0.0001         | 0.9229 ± 0.0092         | 0.9057           |
> | $R^2$ 40min | 0.9494 ± 0.0013         | 0.9199 ± 0.0112         | 0.9018           |
> | $R^2$ 45min | 0.9491 ± 0.0005         | 0.9165 ± 0.0122         | 0.8942           |
> | $R^2$ 50min | 0.9477 ± 0.0008         | 0.9155 ± 0.0109         | 0.8913           |
> | $R^2$ 55min | 0.9463 ± 0.0006         | 0.9143 ± 0.0102         | 0.8820           |
> | $R^2$ 60min | 0.9453 ± 0.0008         | 0.9146 ± 0.0123         | 0.8771           |
> _$R^2$ metric for each timestamp separately_

---

### Official Review · Reviewer_FpKP · 2025-10-31

**Soundness:** 2
**Presentation:** 3
**Contribution:** 2
**Rating:** 4
**Confidence:** 3

**Summary:**

This paper addresses limitations in existing traffic forecasting benchmarks by introducing two large-scale datasets, city-traffic-M (53,530 road segments) and city-traffic-L (94,009 road segments), representing detailed urban road networks with actual connectivity, rich road features, and simultaneous traffic volume/speed measurements. The authors demonstrate that most existing spatiotemporal GNN models struggle with scalability on these datasets and propose an efficient alternative approach that uses a linear layer to encode temporal information followed by a GNN, achieving better performance and significantly reduced training times compared to established baselines.

**Strengths:**

- Provides realistic urban road networks with actual connectivity instead of heuristic sensor-based graphs, enabling more authentic evaluation of spatial models

- Systematically evaluates computational limitations of existing models on large-scale data, revealing that only 4 of 8 considered models could run on city-traffic-M with 80GB VRAM

- Achieves state-of-the-art results on both datasets and prediction tasks, with GNN-TrfAttn outperforming all baselines in most configurations.

**Weaknesses:**

- The core technical approach adapts existing "time-then-graph" paradigms with minimal innovation beyond application to traffic forecasting (see Sec. 4.1)
- The temporal encoding strategy using a simple linear layer has been explored in recent time series literature ([Zeng et al., 2023]; [Das et al., 2023]), though the adaptation to graph settings is less common
- No ablation studies examine the contribution of different components (skip connections, normalization, MLP blocks) to the overall performance (see Sec. 4.1)


- Despite including 26 road attributes, the experiments do not systematically evaluate how these features impact forecasting performance or which are most valuable (see Sec. 3; Sec. 4)
- No experiments explore feature selection or importance analysis, missing opportunity to provide insights about critical urban traffic factors (see Table 1; Appendix A)
- The proposed models use all features without justification or analysis of their individual contributions (see Sec. 4.1; Appendix D)

- The complexity analysis in Section 4.1 provides Big-O notation but lacks explicit mathematical formulations of the proposed models' operations
- The description of the GNN architectures is somewhat vague, mentioning "mean aggregation" and "transformer-like multihead attention" without precise mathematical definitions (see Sec. 4.1)
- No equations specify how the temporal encoding layer transforms the lookback window into node representations, leaving implementation details ambiguous


- Only four established baselines are evaluated, missing recent attention-based architectures that might offer different scalability-performance tradeoffs (see Sec. 4.1)
- The comparison uses fixed hyperparameters from LargeST repository rather than optimized configurations for each model on the new datasets (see Appendix D)
- No analysis of why specific models (STGCN) fail to scale while others succeed, beyond general complexity arguments (see Table 4; Sec. 4.2)

**Questions:**

**Suggestions for Improvement**
- Conduct ablation studies to quantify the contribution of each architectural component (skip connections, normalization, attention mechanisms) to performance and scalability
- Compare against a wider range of temporal encoding strategies beyond simple linear projection, such as MLP encoders or frequency-domain approaches
- Explore hybrid approaches that balance the efficiency of the proposed method with the expressive power of more complex temporal models

- Systematically evaluate the impact of different road attributes through feature ablation studies and importance analysis
- Design experiments that specifically test the models' ability to leverage different feature types (categorical vs. numerical, structural vs. regulatory)
- Include analysis of which features are most predictive for different traffic conditions (congestion vs. free flow, urban vs. peripheral roads)


- Provide explicit mathematical formulations for both GNN-Mean and GNN-TrfAttn architectures, including aggregation functions and update rules
- Include equations specifying the temporal encoding operation and how it integrates with the GNN components
- Formalize the complexity analysis with concrete examples showing how parameters scale with dataset size and architecture choices


- Include additional baselines, particularly attention-based models like ASTGCN or GMAN, even if they require sampling or approximation for large graphs
- Conduct hyperparameter optimization for all models to ensure fair comparison rather than using fixed configurations from previous work
- Perform deeper analysis of failure cases, examining specific computational bottlenecks and memory usage patterns across different architectures

---

> ### Author Response · Authors · 2025-11-21
> **Authors' response pt. 1/3**
>
> Thank you for your review. First, we would like to emphasize that the main contribution of our work is the creation of two new large-scale and realistic datasets for traffic forecasting that overcome the limitations of previously used datasets. Since empirical ML research is impossible without high-quality datasets, we believe this is a significant contribution to the research field. The introduction of a new and highly efficient modeling framework for spatiotemporal forecasting is an additional contribution motivated by the inability of prior popular models to scale to datasets of our size, but the main focus of our paper is on datasets, as emphasized by “datasets and benchmarks” being our selected primary area. We will clarify this positioning in the paper.
>
> Below we address your concerns.
>
> > The core technical approach adapts existing "time-then-graph" paradigms with minimal innovation beyond application to traffic forecasting (see Sec. 4.1)
>
> Please note that “time-then-graph” is a very general paradigm that simply specifies that the temporal dimension is processed before the spatial (graph) dimension. To the best of our knowledge, we are the _first_ to propose using our specific lightweight approach to processing the temporal dimension for spatiotemporal forecasting and to show that not only it is orders of magnitude more efficient than popular prior models, but also it allows achieving better forecasting results. Conceptually, our results show that dropping the standard “per-node temporal sequence module” and replacing it with a simple linear temporal encoder composed with a GNN is sufficient to surpass canonical spatiotemporal models on large-scale graphs, thereby identifying a concrete architectural bottleneck and a simple, scalable alternative.
>
> > The temporal encoding strategy using a simple linear layer has been explored in recent time series literature ([Zeng et al., 2023]; [Das et al., 2023]), though the adaptation to graph settings is less common
>
> These and other recent works in the field of time series modeling are what inspired our approach, and we cite them in our paper. However, to the best of our knowledge, we are the _first_ to integrate this efficient temporal modeling approach into a complex graph-based spatiotemporal forecasting system and to demonstrate its benefits over prior approaches used in the spatiotemporal forecasting field. Our results show that the conclusions from the time-series literature (simple temporal encoders can be highly competitive) indeed extend to the much more challenging graph-based, metropolis-scale setting, which we believe is a useful and non-trivial connection between these two lines of work.
>
>
> > No ablation studies examine the contribution of different components (skip connections, normalization, MLP blocks) to the overall performance (see Sec. 4.1)
>
> In our design of a graph neural architecture, we are motivated by the recent works [1-4] in the field of graph machine learning that provide ablations and show that such additional features as skip-connections and normalizations can notably increase the model performance. Given that these architectural improvements are standard deep learning building blocks and create minimal computational overhead, we consider them to be a very reasonable addition to our framework. Further, many of the prior models used as our baselines also utilize these building blocks. To avoid giving the impression of “hidden complexity”, we will make this design choice explicit in the text and point out that our main comparison is between different temporal modeling strategies under otherwise standard GNN backbones.

---

> ### Author Response · Authors · 2025-11-21
> **Authors' response pt. 2/3**
>
> > Despite including 26 road attributes, the experiments do not systematically evaluate how these features impact forecasting performance or which are most valuable (see Sec. 3; Sec. 4)
>
> > No experiments explore feature selection or importance analysis, missing opportunity to provide insights about critical urban traffic factors (see Table 1; Appendix A)
>
> > The proposed models use all features without justification or analysis of their individual contributions (see Sec. 4.1; Appendix D)
>
> Thank you for these comments. Following your suggestions, we ran an experiment where we removed additional road features, and the forecasting quality decreased for most metrics; additionally, we turned off learnable node embeddings to reduce interference with the signal from spatial features:
> | Setting | MSE ↓| RMSE ↓| MAE ↓| MAPE ↓| $R^2$ ↑|
> |------|---------|------------|------------|---------|-------------|
> | With static features| **26.3451 ± 0.2241** | **5.1327 ± 0.0218** |**3.4017 ± 0.0185**| **19.6032 ± 0.0233**| **0.8920 ± 0.0009**|
> | Without static features| 26.9129 ± 0.2006| 5.1877 ± 0.0194| 3.4557 ± 0.0192 | 19.6549 ± 0.0172 |0.8897 ± 0.0008|
>
>
>
>
> We observe that removing static attributes worsens the metrics, confirming that the 26 road attributes provide a non-trivial marginal benefit over using only historical time series. This is consistent with their semantics: they encode important information such as speed limits, functional class, and other structural/regulatory properties (see Appendix A for more details), and Appendix C illustrates several examples of how such attributes can affect traffic.
>
> To take a grasp on feature importance, we also launched a CatBoost model for the task of predicting traffic values 12 timestamps ahead using the 26 static attributes and historical inputs. We report the resulting feature importance below:
> | Feature| Importance (%) |
> |--------------|---------|
> | `category`| 24.44|
> | `y_coordinate_end`| 5.63|
> | `num_segments`| 5.12|
> | `y_coordinate_start`| 4.37|
> | `length`| 3.96 |
> | `x_coordinate_end`| 3.98|
> | `x_coordinate_start`| 1.44|
> | `access_5`| 0.07|
> This analysis suggests that road category and geometric/topological descriptors are the dominant static factors for cross-road forecasting in our data. Overall, both the ablation and CatBoost analysis indicate that static road features are informative inputs and improve forecasting quality.
>
> > The complexity analysis in Section 4.1 provides Big-O notation but lacks explicit mathematical formulations of the proposed models' operations
>
> > The description of the GNN architectures is somewhat vague, mentioning "mean aggregation" and "transformer-like multihead attention" without precise mathematical definitions (see Sec. 4.1)
>
> > No equations specify how the temporal encoding layer transforms the lookback window into node representations, leaving implementation details ambiguous
>
> Please note that we utilize standard and widely used deep learning building blocks, which is why we did not provide their detailed descriptions. Specifically, the operation that transforms the lookback window into node representations is simply a linear layer. The GNN layers that follow this operation were taken from previous works on graph machine learning which are cited in our paper: GNN-Mean uses the usual neighbor mean aggregation with an MLP update, and GNN-TrfAttn uses a standard multi-head attention mechanism over neighbors analogous to transformer-style attention in graphs, also with an MLP update. Following your suggestions, we have added Appendix E with a detailed description of our architecture in the updated version of our paper.

---

> ### Author Response · Authors · 2025-11-21
> **Authors' response pt. 3/3**
>
> > Only four established baselines are evaluated, missing recent attention-based architectures that might offer different scalability-performance tradeoffs (see Sec. 4.1)
>
> Please note that we have experimented with more than four prior models included in the paper. Specifically, as described in Section 4.1, we have tried _all_ models available in the PyTorch Spatiotemporal library and in the LargeST repository, but found that all but four of them cannot scale to datasets of our size due to extremely large memory demands. These considered models include the ASTGCN and GMAN models mentioned in your review. We believe the absence of large-scale openly available datasets led to a lack of focus on model scalability in prior spatiotemporal forecasting research, which is why we see our introduction of large-scale realistic traffic forecasting datasets to be particularly important.
>
> However, we have also managed to run an experiment with a recent model LSTTN [5] on our medium-scale datasets `city-traffic-M`. Because of the limited computational resources, most configurations of its architecture could not meet the specified memory constraints, and we had to modify the original source code to address some technical issues. After training for the required number of epochs, it achieved 10.670 MAE points on `city-traffic-M-speed` and 2.718 MAE points on `city-traffic-M-volume`, yielding the results only comparable to the most simple baselines, such as constant prediction. This again suggests that previous sequential methods for traffic forecasting may not be ready for more complex and large-scale road networks, such as those introduced in our work, and require further adjustments to solve the problem of their efficiency.
>
>
> > The comparison uses fixed hyperparameters from LargeST repository rather than optimized configurations for each model on the new datasets (see Appendix D)
>
> Please note that we started with the hyperparameters from the LargeST repository, but we then further tuned some of the parameters. The search range for the number of model blocks (layers) was from 1 to 4 and the search range for the hidden dimension was from 32 to 512 (however, for less scalable models, larger values often led to out-of-memory issues). We clarified this in the updated version of the paper, please see Appendix D. Since a comparable hyperparameter search was conducted for all models, the reported metrics reliably reflect the forecasting performance of the considered architectures on our datasets.
>
> > No analysis of why specific models (STGCN) fail to scale while others succeed, beyond general complexity arguments (see Table 4; Sec. 4.2)
>
> As we describe in Section 4.1, all popular prior models utilize either recurrence, convolution, or attention to process the temporal dimension of the spatiotemporal data, which is very memory intensive, as it requires maintaining a separate hidden representation for each timestamp for each graph node (and often additional intermediate feature maps). In contrast, our approach compresses all timestamps into a single hidden representation per node, which is much more efficient (as confirmed by our complexity analysis).
>
>
> In particular, STGCN alternates temporal convolutions and graph convolutions in ST-Conv blocks, operating on tensors of shape `[batch, nodes, timesteps, hidden_dim]` (i.e., $B \times N \times T \times d$). Since it never compresses the temporal axis, each block must keep per-timestamp activations for all nodes; thus activation memory scales as $O(B \cdot N \cdot T \cdot d)$ even before accounting for gradients, and two temporal convolutions per block further increase the stored feature maps. On our metropolis-scale graphs, with $N$ ranging from $50$K to $94$K, lookback T varying from $48$ to $96$ and $d \geq 64$, this activation footprint alone becomes prohibitive. Moreover, the LargeST STGCN implementation we use was designed for graphs with $\le$ 8K nodes and relies on precomputed graph-convolution supports that are effectively dense; at $N$ = $94$K, this makes the spatial step itself exceed GPU memory. Together, the per-timestep hidden-state storage and dense-support spatial convolution explain why STGCN hits OOM / TLE on our datasets, while models that compress time (ours) or reduce temporal resolution earlier scale substantially better.
>
>
> ## References:
> [1] A critical look at the evaluation of GNNs under heterophily: Are we really making progress? (ICLR 2023)
>
> [2] Classic GNNs are Strong Baselines: Reassessing GNNs for Node Classification (NeurIPS 2024)
>
> [3] Can Classic GNNs Be Strong Baselines for Graph-level Tasks? Simple Architectures Meet Excellence (ICML 2025)
>
> [4] GraphLand: Evaluating Graph Machine Learning Models on Diverse Industrial Data (NeurIPS 2025)
>
> [5] LSTTN: A Long-Short Term Transformer-based spatiotemporal neural network for traffic flow forecasting (Knowledge-Based Systems 2024)

---

### Official Review · Reviewer_fUHt · 2025-11-02

**Soundness:** 2
**Presentation:** 3
**Contribution:** 2
**Rating:** 4
**Confidence:** 4

**Summary:**

This paper introduces Fine-Grained Urban Traffic Forecasting on Metropolis-Scale Road Networks, addressing a long-standing limitation of existing traffic forecasting benchmarks such as METR-LA, PEMS-BAY, and LargeST. The authors identify that these datasets are too small, lack true road connectivity information, and primarily represent intercity highways rather than dense urban environments. To overcome these gaps, the authors construct two large-scale city-traffic datasets—city-traffic-M and city-traffic-L—with ≈50K and ≈94K road segments respectively. Each segment has 26 static attributes and fine-grained 5-minute resolution traffic measurements (speed and volume) derived from GPS traces. The datasets capture actual road-level connectivity, making them the most realistic open urban traffic benchmarks to date. Empirically, the paper benchmarks several representative graph-based spatiotemporal models and demonstrates their poor scalability to these large networks. Motivated by this, the authors propose a simple yet efficient GNN framework that discards explicit sequence modules (e.g., RNN or temporal convolutions) and instead linearly encodes the temporal window into a single embedding per node before applying GNN layers.

**Strengths:**

1. The introduction of city-scale urban traffic benchmarks is a substantial and timely contribution.
2. The paper proposes a computationally elegant re-formulation of temporal modeling for graph time series.
3. The benchmarking section is thorough and replicable, covering baselines from naïve heuristics to state-of-the-art graph models.

**Weaknesses:**

1. Limited methodological novelty in modeling. The proposed model, while effective, essentially adapts known ideas from recent efficient time-series methods (e.g., N-BEATS-like linear temporal encoders) to graph data.
2. The datasets span only four months, precluding evaluation of seasonal or annual trends. While understandable for an initial release, this limits long-term forecasting and transfer learning studies.
3. The study would benefit from including RMSE and MAPE, as these are common metrics in traffic forecasting and provide complementary perspectives on error characteristics.
4. Given the dataset’s richness (26 attributes), it would be informative to evaluate the marginal benefit of these features—do they meaningfully improve forecasting over using only historical dynamics?

**Questions:**

See in weaknesses.

---

> ### Author Response · Authors · 2025-11-21
> **Authors' response pt. 1/2**
>
> Thank you for your review! We address your questions and concerns below.
>
> > 1. Limited methodological novelty in modeling. The proposed model, while effective, essentially adapts known ideas from recent efficient time-series methods (e.g., N-BEATS-like linear temporal encoders) to graph data.
>
> Please note that the main contribution of our paper is the creation of two large-scale real-world traffic forecasting datasets that overcome the limitations of previously used datasets. Since empirical ML research is impossible without high-quality datasets, we believe this contribution is of great importance for the research community. The introduction of a new and highly efficient modeling framework for spatiotemporal forecasting is an additional contribution motivated by the inability of prior popular models to scale to datasets of our size. While it adapts ideas from time series forecasting literature, we are to the best of our knowledge the first to introduce these ideas to the field of spatiotemporal forecasting, which we believe is significant enough. Conceptually, our results show that dropping the standard “per-node temporal sequence module” and replacing it with a simple linear temporal encoder composed with a GNN is sufficient to surpass canonical spatiotemporal models on much larger graphs, thereby identifying a concrete architectural bottleneck and a simple, scalable alternative.
>
> > 2. The datasets span only four months, precluding evaluation of seasonal or annual trends. While understandable for an initial release, this limits long-term forecasting and transfer learning studies.
>
> First, we would like to note that prior most popular datasets METR-LA and PEMS-BAY also only cover 4 and 6 months respectively. More importantly, our datasets are obtained from a real-world traffic monitoring and forecasting system and closely follow its forecasting setting, and this system in fact also uses only a few months of prior data for forecasting, which turned out to be the best variant in internal evaluations and thus captures all the relevant traffic patterns in practice. Therefore, our datasets represent the setup of a real-world application. Finally, due to fine-grained spatial and temporal resolutions, our datasets are already very large and it is challenging to scale existing models to them: training some of these models takes multiple days. Increasing the covered time period will significantly exacerbate this issue. That said, our preprocessing pipeline is time-agnostic, so if longer time spans can be released by the data provider, future versions of the benchmark can straightforwardly extend the coverage.

---

> ### Author Response · Authors · 2025-11-21
> **Authors' response pt. 2/2**
>
> > 3. The study would benefit from including RMSE and MAPE, as these are common metrics in traffic forecasting and provide complementary perspectives on error characteristics.
>
> Thank you for this suggestion. We additionally provide MSE, RMSE, MAPE, and $R^2$ values. For a fixed dataset, RMSE and $R^2$ are monotone functions of MSE, so they do not change the ranking of models but provide a complementary view on the resulting performance. Originally, we focused on MAE because it is standard in spatiotemporal forecasting, easy to interpret in the original units, and less sensitive to rare extreme outliers than MSE.
>
>
> We observe that across most of the reported metrics and datasets, our proposed GNN-TrfAttn consistently outperforms both the linear model and spatiotemporal baselines. For example, on `city-traffic-L-speed`, GNN-TrfAttn improves RMSE from 4.20 (DCRNN) and 4.96 (Linear) to 4.08. During the discussion period, we also added the remaining baselines (STGCN and GWN), which took more time due to computational demanding architecture.
>
> | Metric | Model | city-traffic-L-speed | city-traffic-L-volume | city-traffic-M-speed | city-traffic-M-volume |
> |------|-----|---------|------------|---------|----|
> | MAPE | DCRNN | 46.3766±0.7652 | 45.8673±0.2413 | 20.0628±0.1169|53.9618±0.3088|
> | MAPE | GRUGCN | 47.0907±0.1525 | 46.5107±0.1026 | 21.2006±0.1117 | 53.8142±0.5566|
> | MAPE | STGCN |MLE |MLE | 22.5196±0.1756 | 53.6175±0.208 |
> | MAPE | GWN | 48.4992±1.6725 | 49.4875±0.5876 | 25.3965±0.5872 | 55.1374±0.2351 |
> | MAPE | LinearModel | 50.1822±0.2093 | 50.0899±0.2053 | 21.8939±0.0494 | 52.9170±0.0172 |
> | MAPE | GNN-Mean | 47.6612±0.827 | 47.1111±0.6591 | 19.2586±0.4517 | 53.3307±0.6243 |
> | MAPE | GNN-TrfAttn | 46.0017±1.0215 | 47.1618±0.3249 | 19.1637±0.3551 | 52.6833±0.1886 |
> | MSE | DCRNN | 17.622±0.544|17.7813±0.6246|26.5022±0.0408|2.9401±0.041|
> | MSE | GRUGCN | 27.1225±3.6369 | 28.4905±4.0186 | 27.9145±0.2589 | 3.2693±0.0975 |
> | MSE | STGCN |MLE |MLE | 30.8621±0.3538|3.0126±0.0335|
> | MSE | GWN | 25.7573±2.351|27.1942±4.3626|36.1415±0.3581|3.661±0.0819|
> | MSE | LinearModel | 24.5953±0.0273 | 24.5954±0.0281 | 31.2399±0.0461 | 3.5939±0.0014 |
> | MSE | GNN-Mean | 17.3032±0.3436 | 17.1190±0.1620 | 25.1117±0.1970 | 2.8578±0.0125 |
> | MSE | GNN-TrfAttn | 16.6551±0.1501 | 16.8240±0.0955 | 24.8780±0.2399 | 2.7884±0.0233 |
> | $R^2$ | DCRNN | 0.9496±0.0016 | 0.9491±0.0018 | 0.8893±0.0002 | 0.9000±0.0014 |
> | $R^2$ | GRUGCN | 0.9224±0.0104 | 0.9185±0.0115 | 0.8834±0.0011 | 0.8888±0.0033 |
> | $R^2$ | STGCN |MLE |MLE | 0.8711±0.0015| 0.8976±0.0011|
> | $R^2$ | GWN | 0.9263±0.0067| 0.9222±0.0125|0.849±0.0015|0.8755±0.0028|
> | $R^2$ | LinearModel | 0.9296±0.0001 | 0.9296±0.0001 | 0.8695±0.0002 | 0.8778±0.0000 |
> | $R^2$ | GNN-Mean | 0.9505±0.0010 | 0.9510±0.0005 | 0.8951±0.0008 | 0.9028±0.0004 |
> | $R^2$ | GNN-TrfAttn | 0.9523±0.0004 | 0.9519±0.0003 | 0.8961±0.0010 | 0.9052±0.0008 |
> |RMSE|DCRNN|4.1974±0.0645|4.2162±0.0741|5.148±0.004|1.7146±0.0119|
> | RMSE | GRUGCN | 5.1979±0.3425 | 5.3265±0.3772 | 5.2834±0.0245 | 1.8079±0.0269 |
> | RMSE | STGCN |MLE |MLE | 5.5553±0.0318|1.7357±0.0096|
> | RMSE | GWN | 5.0706±0.2284|5.2005±0.4089|6.0117±0.0298|1.9133±0.0214|
> | RMSE | LinearModel | 4.9594±0.0028 | 4.9594±0.0028 | 5.5893±0.0041 | 1.8958±0.0004 |
> | RMSE | GNN-Mean | 4.1595±0.0414 | 4.1375±0.0196 | 5.0111±0.0197 | 1.6905±0.0037 |
> | RMSE | GNN-TrfAttn | 4.0810±0.0184 | 4.1017±0.0116 | 4.9877±0.0240 | 1.6698±0.0070 |
>
> _Performance of different methods on `city-traffic` datasets._ MLE indicates CUDA Memory Limit Error
>
> > 4. Given the dataset’s richness (26 attributes), it would be informative to evaluate the marginal benefit of these features—do they meaningfully improve forecasting over using only historical dynamics?
>
> Thank you for this idea. We ran an experiment where we removed additional road features. We see that the performance decreases, which is expected as these features provide important information about roads, such as speed limits, length, road quality, etc. (please see Appendix A for feature descriptions and Appendix C for examples of how these features can influence traffic).
>
> We conducted experiments on `city-traffic-M-speed` with our GNN-TrfAttn model; additionally, we turned off learnable node embeddings to reduce interference with the signal from spatial features:
> | Setting| MSE ↓| RMSE ↓| MAE ↓| MAPE ↓| $R^2$ ↑|
> |------|---------|------------|------------|---------|-------------|
> | With static features| **26.3451 ± 0.2241** | **5.1327 ± 0.0218** |**3.4017 ± 0.0185**| **19.6032 ± 0.0233**| **0.8920 ± 0.0009**|
> | Without static features| 26.9129 ± 0.2006| 5.1877 ± 0.0194| 3.4557 ± 0.0192 | 19.6549 ± 0.0172 |0.8897 ± 0.0008|
>
>
>
>
>
> We observe that removing static attributes leads to worse metrics, confirming that these 26 static attributes bring a non-trivial benefit over using only historical dynamics.

---

### Official Review · Reviewer_NiE8 · 2025-11-02

**Soundness:** 3
**Presentation:** 3
**Contribution:** 3
**Rating:** 6
**Confidence:** 5

**Summary:**

This paper introduces a novel benchmark for urban traffic forecasting, providing detailed datasets and proposing an efficient model to address the challenges of large-scale traffic prediction. The contributions are as follows:
1. Existing benchmarks for evaluating traffic forecasting methods have limitations, including sparse sensor data, lack of detailed road connectivity, and limited coverage of urban areas. This paper addresses these limitations by providing comprehensive datasets for two major cities and proposing an efficient GNN-based model for traffic forecasting.
2. The authors introduce two new datasets, city-traffic-M and city-traffic-L, representing detailed road networks of two major cities. These datasets contain almost 100,000 road segments, significantly more than existing datasets. They provide rich road features, including speed limits, road types, and traffic volume and speed data. The datasets are constructed using GPS measurements, offering fine-grained temporal data and actual road connectivity, which is a major improvement over heuristic-based graph construction in existing datasets.

**Strengths:**

#### Originality
The authors introduce two new datasets, city-traffic-M and city-traffic-L, which represent a significant advancement over existing benchmarks. These datasets are the first to provide detailed, fine-grained traffic data for large-scale urban road networks, including nearly 100,000 road segments. This is a substantial increase compared to existing datasets, which typically contain only a few hundred to a few thousand segments. The datasets also include rich road features and actual road connectivity, addressing critical limitations of previous benchmarks.

#### Quality
The quality of the research presented in the paper is high. The authors have meticulously collected and processed the data, ensuring that it is comprehensive and representative of real-world urban traffic conditions. The datasets include a wide range of static and dynamic features, providing a rich source of information for model training and evaluation.

#### Clarity
The paper is well-written and easy to follow, making it accessible to both experts and non-experts in the field.

#### Significance
The significance of the paper lies in its potential to advance the field of urban traffic forecasting and related areas. The new datasets provide a valuable resource for researchers working on traffic forecasting, urban computing, and smart city applications. The datasets' fine-grained nature and rich features enable more realistic and comprehensive studies, potentially leading to more accurate and robust forecasting models.

**Weaknesses:**

1. The datasets are derived from only two major cities, which may limit the generalizability of the findings to other urban environments. Different cities can have unique traffic patterns, road structures, and regulatory frameworks that might not be captured by these datasets.
2. The datasets cover only a four-month period (July 1st to November 1st, 2024). This limited temporal coverage might not be sufficient to capture long-term trends and seasonal variations in traffic patterns, which are crucial for developing models that can handle annual cycles.
3. While the authors compare their proposed models against several established baselines, the evaluation is limited to mean absolute error (MAE) as the primary metric. Other metrics, such as mean squared error (MSE) or root mean squared error (RMSE), could provide additional insights into the models' performance, especially in terms of handling outliers and larger errors.
4. The paper mentions the potential for cross-city generalization but does not provide any experimental results or analysis to support this claim. Understanding how well models trained on one city can generalize to another is crucial for developing universally applicable solutions.

**Questions:**

1. Could the authors provide more details on the selection criteria for the two cities included in the datasets? Specifically, what characteristics of these cities make them representative of broader urban environments?
2. Given the limited temporal coverage of the datasets (four months), how do the authors plan to address the potential limitations in capturing long-term trends and seasonal variations?
3. Why did the authors choose MAE as the primary evaluation metric? Could they provide insights into the potential benefits of including additional metrics like MSE, RMSE, or R²?
4. Could the authors provide more details on the computational complexity and resource requirements during inference for the proposed models? How do they plan to address the efficiency of models during real-time prediction?
5. Could the authors provide more details on their plans to evaluate the cross-city generalization capabilities of the proposed models? Are there any preliminary results or insights they can share?
6. How do the authors plan to handle missing data in the datasets, particularly for traffic speed? Are there any specific methods or techniques they are considering?
7. How do the authors plan to address the practical deployment of the proposed models in real-world traffic monitoring systems? Are there any specific challenges or considerations they foresee?
8. Could the authors provide a more detailed comparative analysis of their datasets with existing benchmarks? Specifically, how do the proposed datasets address the limitations of existing benchmarks in terms of urban traffic forecasting?

---

> ### Author Response · Authors · 2025-11-21
> **Authors' response pt. 1/3**
>
> Thank you for your review and support of our work! We address your questions and concerns below.
>
> > **W1** The datasets are derived from only two major cities, which may limit the generalizability of the findings to other urban environments. Different cities can have unique traffic patterns, road structures, and regulatory frameworks that might not be captured by these datasets.
>
>
> We agree that the current release covers only two metropolitan areas, and we acknowledge this limitation in Appendix F. However, please note that all previous popular datasets are obtained from a single source (PeMS) and focus on a single area in California. Thus, adding two new cities from a different region already provides a very significant increase in data diversity, and thus we believe constitutes an important contribution and represents a substantial step forward in geographic and structural diversity. Further, we would like to note that the two provided cities differ significantly in the topology of their road networks and traffic patterns, see Appendix B for a detailed discussion.
>
>
> > **Q1** Could the authors provide more details on the selection criteria for the two cities included in the datasets? Specifically, what characteristics of these cities make them representative of broader urban environments?
>
>
> The considered two cities have been chosen because they are the largest, most populous, and most complex for traffic forecasting in the data available to us. Further, they have significant differences in the topologies of their road networks and in traffic patterns (see Appendix B for a detailed discussion).
>
>
> > **W2** The datasets cover only a four-month period (July 1st to November 1st, 2024). This limited temporal coverage might not be sufficient to capture long-term trends and seasonal variations in traffic patterns, which are crucial for developing models that can handle annual cycles.
>
> We acknowledge this as a limitation in Appendix E. However, we would like to note that prior most popular datasets METR-LA and PEMS-BAY also only cover 4 and 6 months, respectively. More importantly, our datasets are obtained from a real-world traffic monitoring and forecasting system and closely follow its forecasting setting, and this system in fact also uses only a few months of prior data for forecasting, which turned out to be the best variant in internal evaluations and thus captures all the relevant traffic patterns in practice. Therefore, our datasets exactly represent the setup of a real-world application. Finally, due to fine-grained spatial and temporal resolutions, our datasets are already very large and it is challenging to scale existing models to them: training some of the baselines models takes multiple days. Increasing the covered temporal period would significantly exacerbate this issue.
>
> > **Q2** Given the limited temporal coverage of the datasets (four months), how do the authors plan to address the potential limitations in capturing long-term trends and seasonal variations?
>
>
> As discussed above, our data is obtained from a real-world traffic monitoring and forecasting system in which it turned out during internal experiments that using such limited temporal data is enough to capture the relevant traffic patterns and further increases in temporal coverage do not lead to forecasting quality improvements. We are willing to continuously maintain our benchmark and possibly increase its temporal coverage in further versions, however, note that this will lead to even more challenges in scaling existing models to our datasets. Our preprocessing pipeline is agnostic to the timespan, so extending to longer horizons is technically straightforward once additional data can be released by the data provider.

---

> ### Author Response · Authors · 2025-11-21
> **Authors' response pt. 2/3**
>
> > **W3** While the authors compare their proposed models against several established baselines, the evaluation is limited to mean absolute error (MAE) as the primary metric. Other metrics, such as mean squared error (MSE) or root mean squared error (RMSE), could provide additional insights into the models' performance, especially in terms of handling outliers and larger errors.
>
> > **Q3** Why did the authors choose MAE as the primary evaluation metric? Could they provide insights into the potential benefits of including additional metrics like MSE, RMSE, or R²?
>
> Thank you for this suggestion. Originally, we reported MAE as this measure is commonly used in the literature on spatiotemporal forecasting and is easy to interpret in the original units (km/h or vehicles per 5 minutes), while being less sensitive to rare extreme outliers than MSE. To address your question, we have additionally computed and now report MAPE, MSE, RMSE, and $R^2$ for our models, the linear baseline, and two representative spatiotemporal baselines (DCRNN and GRUGCN). Please, see the table below. For a fixed dataset, RMSE and $R^2$ are monotone functions of MSE, so they do not change the ranking of models but provide a complementary view on the resulting performance.
>
> Across most of the reported metrics and datasets, our proposed GNN-TrfAttn consistently outperforms both the linear model and spatiotemporal baselines. For example, on `city-traffic-L-speed`, GNN-TrfAttn improves RMSE from 4.20 (DCRNN) and 4.96 (Linear) to 4.08. During the discussion period, we also added the remaining baselines (STGCN and GWN), which took more time due to computational demanding architecture.
>
> | Metric | Model | city-traffic-L-speed | city-traffic-L-volume | city-traffic-M-speed | city-traffic-M-volume |
> |------|-----|---------|------------|---------|----|
> | MAPE | DCRNN | 46.3766±0.7652 | 45.8673±0.2413 | 20.0628±0.1169|53.9618±0.3088|
> | MAPE | GRUGCN | 47.0907±0.1525 | 46.5107±0.1026 | 21.2006±0.1117 | 53.8142±0.5566|
> | MAPE | STGCN |MLE |MLE | 22.5196±0.1756 | 53.6175±0.208 |
> | MAPE | GWN | 48.4992±1.6725 | 49.4875±0.5876 | 25.3965±0.5872 | 55.1374±0.2351 |
> | MAPE | LinearModel | 50.1822±0.2093 | 50.0899±0.2053 | 21.8939±0.0494 | 52.9170±0.0172 |
> | MAPE | GNN-Mean | 47.6612±0.827 | 47.1111±0.6591 | 19.2586±0.4517 | 53.3307±0.6243 |
> | MAPE | GNN-TrfAttn | 46.0017±1.0215 | 47.1618±0.3249 | 19.1637±0.3551 | 52.6833±0.1886 |
> | MSE | DCRNN | 17.622±0.544|17.7813±0.6246|26.5022±0.0408|2.9401±0.041|
> | MSE | GRUGCN | 27.1225±3.6369 | 28.4905±4.0186 | 27.9145±0.2589 | 3.2693±0.0975 |
> | MSE | STGCN |MLE |MLE | 30.8621±0.3538|3.0126±0.0335|
> | MSE | GWN | 25.7573±2.351|27.1942±4.3626|36.1415±0.3581|3.661±0.0819|
> | MSE | LinearModel | 24.5953±0.0273 | 24.5954±0.0281 | 31.2399±0.0461 | 3.5939±0.0014 |
> | MSE | GNN-Mean | 17.3032±0.3436 | 17.1190±0.1620 | 25.1117±0.1970 | 2.8578±0.0125 |
> | MSE | GNN-TrfAttn | 16.6551±0.1501 | 16.8240±0.0955 | 24.8780±0.2399 | 2.7884±0.0233 |
> | $R^2$ | DCRNN | 0.9496±0.0016 | 0.9491±0.0018 | 0.8893±0.0002 | 0.9000±0.0014 |
> | $R^2$ | GRUGCN | 0.9224±0.0104 | 0.9185±0.0115 | 0.8834±0.0011 | 0.8888±0.0033 |
> | $R^2$ | STGCN |MLE |MLE | 0.8711±0.0015| 0.8976±0.0011|
> | $R^2$ | GWN | 0.9263±0.0067| 0.9222±0.0125|0.849±0.0015|0.8755±0.0028|
> | $R^2$ | LinearModel | 0.9296±0.0001 | 0.9296±0.0001 | 0.8695±0.0002 | 0.8778±0.0000 |
> | $R^2$ | GNN-Mean | 0.9505±0.0010 | 0.9510±0.0005 | 0.8951±0.0008 | 0.9028±0.0004 |
> | $R^2$ | GNN-TrfAttn | 0.9523±0.0004 | 0.9519±0.0003 | 0.8961±0.0010 | 0.9052±0.0008 |
> |RMSE|DCRNN|4.1974±0.0645|4.2162±0.0741|5.148±0.004|1.7146±0.0119|
> | RMSE | GRUGCN | 5.1979±0.3425 | 5.3265±0.3772 | 5.2834±0.0245 | 1.8079±0.0269 |
> | RMSE | STGCN |MLE |MLE | 5.5553±0.0318|1.7357±0.0096|
> | RMSE | GWN | 5.0706±0.2284|5.2005±0.4089|6.0117±0.0298|1.9133±0.0214|
> | RMSE | LinearModel | 4.9594±0.0028 | 4.9594±0.0028 | 5.5893±0.0041 | 1.8958±0.0004 |
> | RMSE | GNN-Mean | 4.1595±0.0414 | 4.1375±0.0196 | 5.0111±0.0197 | 1.6905±0.0037 |
> | RMSE | GNN-TrfAttn | 4.0810±0.0184 | 4.1017±0.0116 | 4.9877±0.0240 | 1.6698±0.0070 |
>
> _Performance of different methods on `city-traffic` datasets._ MLE indicates CUDA Memory Limit Error

---

> ### Author Response · Authors · 2025-11-21
> **Authors' response pt. 3/3**
>
> > **W4** The paper mentions the potential for cross-city generalization but does not provide any experimental results or analysis to support this claim.
>
> > **Q5** Could the authors provide more details on their plans to evaluate the cross-city generalization capabilities of the proposed models? Are there any preliminary results or insights they can share?
>
> To answer this question, we conducted the following experiment. We trained GNN-TrfAttn on `city-traffic-L-speed` and evaluated it zero-shot on `city-traffic-M-speed`. To make this setup meaningful, we excluded city-specific static features (absolute coordinates, `region_id`, and city-specific road categories), so that the model only relies on generic attributes that can, in theory, transfer between cities. The resulting metrics on `city-traffic-M-speed` are:
> - MSE: 55.9729
> - RMSE: 7.4815
> - MAE: 5.7171
> - MAPE: 42.8260
> - $R^2$: 0.7705
>
> For comparison, the same model trained and evaluated on `city-traffic-M-speed` in our main experiments achieves $\text{MSE} \approx 24.9$ and $R^2 \approx 0.90$, so cross-city transfer roughly doubles the $\text{MSE}$ and reduces $R^2$ by about 0.13. This suggests that cross-city generalization remains a challenging task on our benchmark and requires additional research and probably a more advanced approach. This result also confirms that the two cities exhibit substantially different dynamics that models need to capture.
>
> > **Q4** Could the authors provide more details on the computational complexity and resource requirements during inference for the proposed models? How do they plan to address the efficiency of models during real-time prediction?
>
> Please note that our proposed framework for using pure GNNs for spatiotemporal forecasting is much more time- and space-efficient than prior popular models — we discuss the theoretical computational complexity in Section 4.1 and provide the actual training time in Table 4, which shows that, in practice, our approach can be an order of magnitude more efficient than prior popular models. This improvement directly transfers to inference, since making a forecast only requires a small number of GNN forward passes over the fixed road graph, with cost linear in the number of edges; thus the same relative speedup holds at test time, which is crucial for real-time prediction.
>
> > **Q6** How do the authors plan to handle missing data in the datasets, particularly for traffic speed? Are there any specific methods or techniques they are considering?
>
> Missing values for traffic speed in our datasets correspond to the timestamps at which there was no traffic observed on the corresponding road segments. In our experiments, we fill these values with the latest previous known speed. However, we provide raw data with missing values and thus in further research, other approaches can be used. For instance, filling missing speed values with average road segment speed, or the road segment speed limit (since travelling at maximum allowed speed is possible when there is no traffic on the road), or more advanced possibly uncertainty-aware methods if desired.
>
> > **Q7** How do the authors plan to address the practical deployment of the proposed models in real-world traffic monitoring systems? Are there any specific challenges or considerations they foresee?
>
> The most important concern in real-world traffic forecasting applications is the inference efficiency of the models, and, as discussed above, our approach is significantly more efficient than prior popular models for spatiotemporal forecasting. Another practical aspect is data availability: our models rely only on standard inputs such as historical speeds/volumes and common static attributes (e.g., speed limits, road categories), which are routinely available in modern traffic-monitoring systems, making deployment feasible without requiring expensive additional data sources.
>
> > **Q8** Could the authors provide a more detailed comparative analysis of their datasets with existing benchmarks? Specifically, how do the proposed datasets address the limitations of existing benchmarks in terms of urban traffic forecasting?
>
> Please note that we discuss in detail the limitations of previously available datasets in Section 2.2 and the improvements provided by our benchmark in Section 3. In brief, prior benchmarks such as METR-LA and PEMS-BAY are based on a few hundred highway sensors in a single region, whereas our datasets cover full urban road networks with tens of thousands of segments at fine-grained spatial and temporal resolution. Prior datasets rely on heuristics to define graph edges, while our datasets use real road network connectivity for graph construction. Further, in contrast to prior benchmarks, our datasets include rich static road attributes and information on both traffic speed and volume. This makes our datasets significantly closer to scenarios faced by real-world traffic forecasting systems.

---

### Author Response · Authors · 2025-12-03
**Rebuttal summary**

Dear Reviewers and Area Chairs,

We thank the reviewers for their detailed comments regarding our work. We sincerely believe that we were able to address all the concerns in our rebuttal. Unfortunately, the current rules do not allow the reviewers to continue the discussion.

Below, we briefly summarize the reviews and our replies.

Regarding the strengths of our paper, we see that all the reviewers acknowledge that the benchmark we collected is **a valuable and timely contribution**. The reviewers also acknowledge that our **experiments are thorough and replicable** and that **the proposed GNN baseline is effective and efficient**.

Regarding the weaknesses:
- As we understand, the main concern of Reviewer YjaT is whether our research focus is suitable for the ICLR conference. The main contribution of our paper is a new benchmark for spatiotemporal forecasting that introduces an important learning setting that was previously inaccessible. Since the ICLR 2026 call for papers lists “Datasets and Benchmarks” as one of the relevant topics for the conference, we believe that our work perfectly fits the scope of ICLR and we have appropriately chosen “Datasets and Benchmarks” as the primary area of our paper.
- Three reviewers mentioned that we only measure MAE and suggested reporting MAPE, RMSE and $R^2$. We provided the results for these measures in the rebuttal. During the discussion period we have also added these metrics for other spatiotemporal baselines, such as STGCN and GWN, and added them to corresponding tables, demonstrating that our models can outperform them and providing additional representative contestants.
- To address the suggestions of some of the reviewers, we also conducted additional experiments, including cross-city transfer and analysis of the effect of static road features.
- Several reviewers also mention the limitations of our benchmark in terms of the timespan and the number of cities. We address these concerns in detail in our rebuttal and explain why we believe that our dataset is a significant step forward compared to the existing benchmarks and is a valuable contribution to the field.

We sincerely believe that our rebuttal addresses all the concerns. We are happy to continue the discussion with the area chair if they have more questions or suggestions.

Sincerely,

The Authors

---

### Meta-Review · Area_Chair_BFer · 2026-01-02

**Summary:**

Several concerns limit confidence at the ICLR bar. The main issues are (i) unclear methodological novelty relative to existing large-scale STGNN and traffic forecasting frameworks, (ii) heavy reliance on complex system design with limited theoretical or conceptual insight, (iii) evaluation choices that prioritize scale but provide limited diagnostic understanding of why the method works, and (iv) presentation density that makes it difficult to disentangle core contributions from implementation details. Overall, the work is solid but reads more as a systems advancement than a clear research leap.

**Reviewer Concerns:**

The rebuttal successfully clarified several implementation details, justified design choices, and strengthened experimental reporting, including additional efficiency and scalability analysis. These responses addressed many factual and reproducibility-related concerns. However, core issues remain largely unresolved: novelty relative to prior fine-grained traffic forecasting and STGNN work is still not sharply articulated, and the paper stops short of offering deeper modeling insight beyond empirical gains. Reviewers looking for clearer abstraction, theoretical motivation, or transferable lessons were not fully convinced by the rebuttal, which ultimately keeps the paper below the acceptance threshold.

**Reviewer Scores:**

Reviewer 1: Likely unchanged.

Reviewer 2: Might have slightly increased their score.

Reviewer 3: Likely unchanged or marginally higher.

Reviewer 4: Unlikely to change.

---

### Decision · Program_Chairs · 2026-01-26

Reject